# Single-cell transcriptome atlas of the human corpus cavernosum

LiangYu Zhao[1,2,7], Sha Han[1,7], HengChuan Su[3,7], JianYing Li[1,7], ErLei Zhi[1], Peng Li[1], ChenCheng Yao[1], RuHui Tian[1], HuiXing Chen[1], HuiRong Chen[1], JiaQiang Luo[1], ChenKun Shi[1], ZhiYong Ji[1], JianLin Hu[1], Gang Wu[4], WeiDong Zhou[4], YuXin Tang[2], YuZhuo Chen[5], GuiTing Lin[6], Tom F. Lue[6] ✉, DengLong Wu[3] ✉ & Zheng Li [1] ✉

The corpus cavernosum is the most important structure for penile erection, and its dysfunction causes many physiological and psychological problems. However, its cellular heterogeneity and signalling networks at the molecular level are poorly understood because of limited access to samples. Here, we profile 64,993 human cavernosal single-cell transcriptomes from three males with normal erection and five organic erectile dysfunction patients. Cell communication analysis reveals that cavernosal fibroblasts are central to the paracrine signalling network and regulate microenvironmental homeostasis. Combining with immunohistochemical staining, we reveal the cellular heterogeneity and describe a detailed spatial distribution map for each fibroblast, smooth muscle and endothelial subcluster in the corpus cavernosum. Furthermore, comparative analysis and related functional experiments identify candidate regulatory signalling pathways in the pathological process. Our study provides an insight into the human corpus cavernosum microenvironment and a reference for potential erectile dysfunction therapies.

Sexual reproduction and the consequent relationships of couples are important foundations of human society and culture. Sexual health is thought to reflect overall male health; however, discussing this problem is often considered taboo, so less academic attention than other physiological problems has resulted in few studies on the human corpus cavernosum microenvironment. Through evolution, the human penis has lost structures such as the penis bone. Thus, the penis needs to expand several times in size during sexual intercourse, and the bulbocavernosus reflex during erection rigidity causes the intracorporal pressure to exceed the systolic blood pressure[1,2]. During this process, erectile signals are transmitted to the corpora cavernosa

(CC) by activation of the parasympathetic nervous system and signal transfer in the sacral erectile centre, regulating the nitric oxide synthase (NOS)/cGMP cascade, subsequently leading to depolarisation followed by smooth muscle relaxation with consecutively increased blood inflow[3,4]. The CC is the basic functional unit of the penile erection process. However, its microenvironment's constituents remain unclear, especially the fibroblasts (FB). These cells comprise about half of the total cavernosal cells, and research on their effect on the normal physiological function of the CC remains inadequate.

Erectile dysfunction (ED) causes physiological and psychological problems for patients and their families. The CC can be considered as

[1]Department of Andrology, the Center for Men's Health, Urologic Medical Center, Shanghai Key Laboratory of Reproductive Medicine, Shanghai General Hospital, Shanghai Jiao Tong University School of Medicine, Shanghai 200080, China. [2]Department of Urology, Department of Interventional Medicine, Guangdong Provincial Key Laboratory of Biomedical Imaging, The Fifth Affiliated Hospital, Sun Yat-sen University, Zhuhai, Guangdong Province 519000, China. [3]Department of Urology, Fudan University Shanghai Cancer Center, Department of Oncology, Shanghai Medical College, Fudan University, Shanghai 200032, China. [4]Department of Urology, Tongji Hospital, School of Medicine, Tongji University, Shanghai 200065, China. [5]Department of Ultrasound, The Fifth Affiliated Hospital, Sun Yat-sen University, Zhuhai, Guangdong Province 519000, China. [6]Knuppe Molecular Urology Laboratory, Department of Urology, School of Medicine, University of California, San Francisco, CA, USA. [7]These authors contributed equally: LiangYu Zhao, Sha Han, HengChuan Su, JianYing Li. ✉e-mail: tom.lue@ucsf.edu; wudenglong2009@tongji.edu.cn; lizhengboshi@sjtu.edu.cn

a special blood vessel type, and its dysfunction is a harbinger of cardiovascular disease events, indicating that the corpus cavernosum is more susceptible to injury than other blood vessels[5]. Investigational cavernosal injury biomarkers may provide a window for the primary and secondary prevention of ED, and for cardiovascular disease and other vascular disorders[6]. ED's pathogenesis can be further divided into the stages of functional disorder and organic disturbances. Phosphodiesterase type-5 (PDE5) inhibitors increase intracellular cGMP levels in vascular and corporal SMCs and thus potentiates the NO-mediated vasorelaxant response to sexual arousal[7,8]. However, smooth muscle relaxation is only one part of the erectile process. No suitable treatment yet exists for cavernosal structural damage of ED patients at late stages because this depends on a full understanding of the cavernosal microenvironment, including the cell heterogeneity, the regulatory signalling, and the interactions with other cell types at the molecular level. In addition, most previous studies on the CC were based on the rat model, but the human and rat penis structure and erection process show large differences, so the relevant conclusions were sometimes not matched with the observation of human study[9].

To overcome the above problems, we profiled 64,993 individual penile cavernosal cells from three men with normal erections and five patients with organic ED. Based on these data, we developed a single-cell atlas of the human CC and described the communication network within its microenvironment. Seven major cavernosal cell types, including six FB subclusters, four smooth muscle cell (SMC) subclusters, and four endothelial cell (EC) subclusters were identified with distinct transcriptome profiles. Each subcluster's heterogeneity, including the spatial location, biological function, and regulatory pathway, was also identified. Furthermore, according to the distinct transcriptome profiles between normal and ED cavernosal cells, we obtained evidence that the Wnt and Hippo signalling pathways regulate the damage and repair of the CC tissue. Collectively, our results provide an in-depth insight into the cavernosal cellular microenvironment and the mechanisms underlying pathogenesis, thereby offering targets for ED prevention and treatment strategies.

## Results

### Overview of the cellular composition of the human CC

To characterise the CC's cellular heterogeneity and baseline cellular diversity in a pathological state, we profiled CC cells from three donors with normal erections and five ED patients, including two patients with diabetes mellitus erectile dysfunction (DMED) and three patients with non-diabetic erectile dysfunction (non-DM) (Supplementary Data 1). Thus, 64,993 cells were retained after removing empty droplets, outliers, cell debris, and inferential doublets (Fig. 1a and Supplementary Fig. 1a). All donors ranged in age from 40 to 55, and each donor's sex hormones were within the normal range (Supplementary Fig. 1a, b). Pathological examination revealed the five ED patients' cavernosum tissue had obviously undergone fibrosis; the muscle/collagen ratio decreased by 50% in the three non-DM cases and by as much as 70% in the diabetes cases. In addition, the size of the cavernous sinus lumen in ED patients decreased (Supplementary Fig. 1c, d). Cells were further filtered according to the number of genes and reads, and each sample had a similar number of genes expressed (Supplementary Fig. 1e).

Seven major clusters were identified in the whole cell population based on the expression of known cell type-specific markers, including ECs, FBs, pericytes (PCs), macrophages (MACs), SMCs, Schwann cells (SWCs), and T cells (Ts). All clusters contained both normal and ED patient cells, but normal and ED CC showed significant heterogeneity in t-SNE plot (Fig. 1b, c). Each cluster had different gene transcription levels (Supplementary Fig. 1f). These seven clusters could further divided into 11 clusters according to the cells whether belong to cavernosal trabecular or the nearby vessels (Supplementary Fig. 1g). t-SNE plot of these cells showed good repeatability within the three normal samples and same type of ED patients (Supplementary

Fig. 1h). A total of 6148 differentially expressed genes (DEGs) among the seven clusters under normal state were identified (Supplementary Fig. 1i and Supplementary Data 2). Gene ontology (GO) analysis was performed for the top 30 DEGs of each cluster, the results consistent with the previous understanding of the biological characteristics of these cell types (Fig. 1d). Of note, the FB cluster showed the most powerful signal output capacity among the seven clusters (Fig. 1e and Supplementary Fig. 2b, c) indicated their important regulatory function in the CC microenvironment.

### Cell–cell interaction analysis in CC microenvironment under normal and disease state

To investigate the CC microenvironment's complex signalling networks, we performed unbiased "ligand–receptor", "extracellular matrix (ECM)–receptor" and "cell–cell contact" interaction analyses by Cell-Chat, and identified 1284, 1461 and 1552 significant interactions in normal, non-DM and DMED CC respectively (Supplementary Fig. 2a, b). Among the seven clusters, FBs showed the strongest outgoing signalling pattern, indicating that FBs were key in regulating the microenvironment's homeostasis (Supplementary Fig. 2c). Unexpectedly, the MAC, SWC and T clusters showed a typical signal incoming pattern. Among these signalling interactions, we discovered many classical vascular tissue regulatory pathways, such as VEGF, IGF, NOTCH, et al. (Supplementary Fig. 2c)[10,11]. And further identified some signals that showed a different pattern between normal and pathological state such as KIT (receptor of SCF), FGF, EGF, et al. (Supplementary Fig. 2d).

Then, RNA-seq, ICC and ELISA experiments were performed to verify whether these ligands were indeed produced by CC FB at the RNA and protein levels. RNA-seq and ICC of FB indicated most of the ligands predicted by CellChat were expressed in FB in vitro (Supplementary Fig. 3a, c), and RNA-seq of EC indicated their paired receptors were expressed in EC (Supplementary Fig. 3b). The ELISA results showed that EC culture consumed most of the ligand molecules, and FB culture increased most of these factors' concentrations (Supplementary Fig. 3d). These experimental results confirm CellChat's prediction of the expression patterns of these signal ligands.

Next, we evaluated the effects of these signal ligands on EC functions, which were related to the repair of CC tissue after pathological injury. In the single factor variable experiment, IGF1 treatment had a positive effect on tube formation (Supplementary Fig. 4b, d), vascular bud length (Supplementary Fig. 4g), and cell migration (Supplementary Fig. 4j) of EC, while AGEs (Advanced Glycation End Products) and TGFβ acted as a negative regulator. These results were also verified by orthogonal experiments with various variables (Supplementary Fig. 4c, e, h, k). In addition, supernatant of FB also promoted above functions of EC, however, expect for cell migration, these effects disappeared after FB were treated with AGEs.

Nitric oxide (NO) and endothelin (ET) were two important regulators for the contraction and relaxation of SMC. The ELISA results indicated that most signal molecules even TGFβ and low level of AGEs treatment alone showed a positive effect on endothelial NO releasing (Supplementary Fig. 5a). However, the effect of each signal ligand significantly decreased in the orthogonal experiment (Supplementary Fig. 5b and Supplementary Data 6). Both high and low concentrations of AGEs can effectively stimulate the ET releasing. Except for VEGFA and CXCL12, most of the factors show inhibitory effects on ET releasing, especially the HBEGF has shown a strong inhibitory effect in both single factor stimulation and orthogonal experiment (Supplementary Fig. 5c, d, and Supplementary Data 6). Furthermore, we found high concentrations of AGEs, TGFβ, and Hippo pathway inhibitor XMU-MP-1 led to increased endothelial permeability (Supplementary Fig. 5e). TGFβ also showed a similar effect in the orthogonal experiment (Supplementary Fig. 5f), but there was no statistical difference (Supplementary Data 6). In general, IGF1 and SCF signals positively regulated CC tissue repair-related function. In addition,

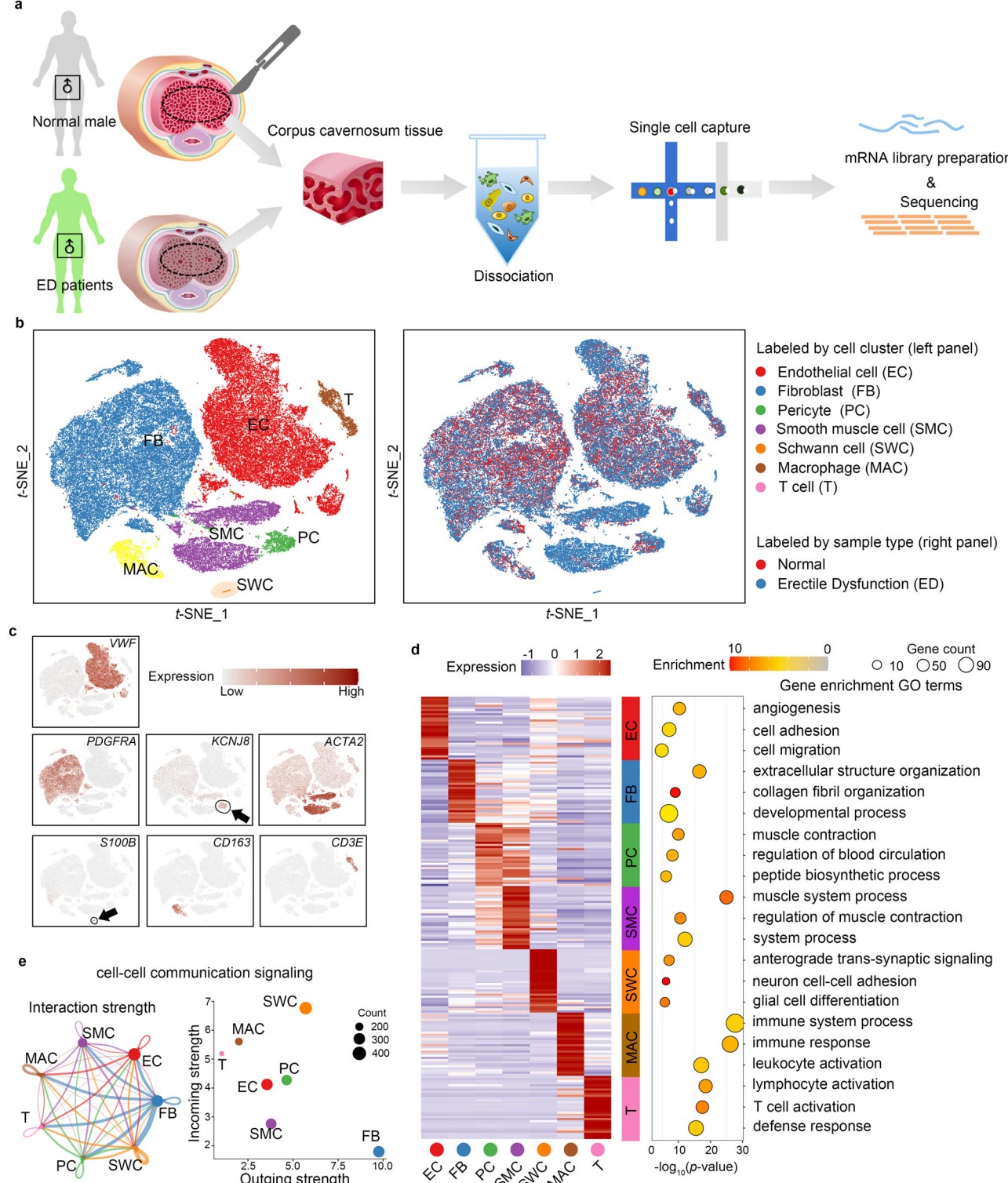

**Fig. 1 | Global expression profiling of human corpus cavernosum cells in normal male and ED patients by single-cell RNA-seq. a** Schematic illustration of the experimental workflow in this study. **b** t-distributed stochastic neighbour embedding (tSNE) plots of all corpus cavernosum cells from eight donors (three normal males and five ED patients). Cells are coloured according to their types (left panel) or state (right panel; red = Normal, blue = Erectile dysfunction). **c** Expression patterns of the following marker genes for each cluster are projected on the tSNE plot of all CC cells: *VWF* (endothelial cells), *PDGFRA* (fibroblast), *KCNJ8* (pricytes), *ACTA2* (smooth muscle cells), *S100B* (Schwann cells), *CD163* (macrophages) and *CD3E* (T cells). A gradient of light grey to dark red indicates low to high expression levels. **d** Heatmap of the top 30 DEGs in each major cluster (left

panel), with the GO analysis (biological process) according to the DEGs of each major cluster shown as a bubble diagram (right panel). A gradient of light blue to dark red indicates low to high expression levels in the heatmap. Statistical analysis was based on Fisher's exact test; two-tailed; the confidence interval is 95%. A gradient of red to grey indicates low to high *P*-values in the bubble diagram and the size of the bubbles indicates the count of enriched DEGs for each GO term. **e** Cell–cell communication signalling network among the seven major clusters analysed with CellChat. The width of the lines indicates the number of pairs. Different colours represent different signal sources. The right panel showed that cell clusters were located based on the count of their significant incoming (Y-axis) or outgoing (X-axis) signalling pattern.

IGF1, SCF, FGF7 and HBEGF signals were beneficial to the regulatory function of smooth muscle relaxation, while high concentrations of AGEs and TGFβ were harmful to these functions. These signalling network results helped in further understanding the cavernosal microenvironment's physiological and pathological processes.

## Cellular heterogeneity of biological processes and spatial localisation in the FB cluster

To further explore the corpus cavernosal FB (CCFB) and their biological function in a framework of fibroblast lineage, we first combined our CCFB with the fibroblast dataset from 9 different studies including heart[12], skeletal muscle[13], colon[14], liver[15], kidney[16], skin[17,18] and lung[19,20] under normal and disease state. The similarity between fibroblasts derived from each tissue was compared and we found that CCFB was more similar to that of heart and skeletal muscle (Supplementary Fig. 6c). After referring to the classification criteria of Buechler's study[21], we found CCFB mainly consists of three clusters, PI16+, APOC1+ and COMP+ FB (Supplementary Fig. 6a, b). PI16 was predicted to act upstream of or within negative regulation of cell growth involved in cardiac muscle cell development and this cluster was thought as a root in trajectories of fibroblast differentiation[21]. These three clusters could further divide into 6 subclusters based on some CCFB specific markers including APOC1+/PTCHD1+ (FB1), APOC1+ /PPP1R14A+ (FB2), PI16+/FMO2+ (FB3), PI16+/BMP7+ (FB4), COMP+/KERA+ (FB5) and COMP+/MFAP5+ FB (FB6) (Fig. 2a). We identified 212, 151, 71, 429, 254 and 442 DEGs among these six CCFB subclusters under normal state, respectively (Fig. 2b, c and Supplementary Data 8). Then the DEGs between normal and ED state in each CCFB subcluster were also calculated (Supplementary Fig. 7 and Supplementary Data 9). We found that muscle-related GO terms, such as "muscle system process" and "muscle cell proliferation" were enriched in the PI16-FB (FB1 and FB2) subclusters. The FB4 cluster containing KERA, THBS4, SCX and collagens was consistent with a tenogenic signature. Although the general ECM genes were expressed at relatively higher levels in FB3, FB4 and FB5, each collagen type showed different expression patterns in the six FB subclusters. For example, collagen I was highly expressed in FB4 and FB5, while collagen II and collagen IV were highly expressed in FB1 and FB2 (Supplementary Fig. 8a). The collagen expression pattern also differed between normal and ED FB. For example, normal FB showed higher *COL1A1* and *COL6A3* expression levels but a lower *COL4A1* expression level compared with FB from ED patients (Supplementary Fig. 8b, c).

We then analysed the spatial localisation of each FB subcluster by immunohistochemical staining (IHC) of the CC sections. We first divided the CC into five major regions: (a) the septum pectiniforme, (b) the region near the septum pectiniforme, (c) the cavernosal trabecular region, (d) the cavernosal artery and (e) the nerve bundles (Supplementary Fig. 9a). *PI16* was highly expressed in FB3, FB5 and FB6, and the IHC results showed that part of PI16+ cells existed in the artery region, while other PI16+ cells located between the smooth muscle bundles in the cavernosal trabecular region (Supplementary Fig. 9b). WISP2 showed a similar expression pattern and spatial localisation as PI16, but *WISP2* was not expressed in FB6 and no WISP2+ cells were detected outside the artery region (Supplementary Fig. 9c). *SRDSA2* was expressed in FB1, FB4 and part of FB3, and the SRD5A2+ cells were located in the region around the septum pectiniforme (Supplementary Fig. 9d). Neurofilament light chain transcript (*NEFL*) was only expressed in FB4, indicating that FB4 was highly correlated with neurons. The IHC results showed that FB4 mainly existed in large nerve bundles near the deep penis artery and in small nerve bundles within the cavernosal trabecular region (Supplementary Fig. 9e). Most NEFL+ cells were coated by a layer of NGFR+ cells that were identified as the FB6 cluster (Supplementary Fig. 9f).

When compared with ED patients, the proportion of the APOC1+ FB, especially FB1 subcluster in both non-DM and DMED patients was increased (Fig. 2d and Supplementary Fig. 6a, b). We then used ingenuity pathway analysis (IPA) to analyse each subcluster's cell death and proliferation and found that proliferation- and survival-related terms, such as "cell cycle progression", "proliferation of epithelial cells", "cell viability" and "cell survival", were significantly activated in the FB1 subcluster (Fig. 2e). Regarding heterogeneity in energy metabolism, we found that FB1 expressed a lower level of glycolysis-related genes, while FB1 and FB2 expressed a higher level of triglyceride metabolic-related genes (Fig. 2f).

## Regulatory networks involved in phenotypic transformation of FBs

Considering that the FB1 subcluster showed muscle-relate characteristics and its proportion had increased in ED patients, we hypothesised that the FB1 subcluster is associated with excessive contraction and fibrosis of the cavernous sinus. To determine which signal pathway regulate phenotypic transformation among FB subclusters, we built a cell interaction network between the FBs and other CC cells with CellChat. The CellChat result indicated that FB1 mainly received WNT, IL17, NT and VCAM signalling (Fig. 3a). Then, we further identified regulatory pathways in each FB subcluster by IPA according to the downstream DEGs. IPA analysis showed that terms such as "Wnt/β-catenin signalling", "VEGFs interactions", "NAD signalling" and "Endothelin-1 signalling" were activated (Fig. 3b). To further explore the key regulators of each FB subcluster, we analysed 1665 human transcription factors and their downstream gene set using GENIE3. We found that factors such as FOS, EGR1 and JUN might specifically play critical roles in the FB1 subcluster (Fig. 3c), and other subclusters also had their own special regulators (Supplementary Fig. 11a). Interestingly, among these transcription factors, JUN and TBX3 are direct targets of Wnt/β-catenin signalling, further demonstrating its importance. Then, we focused on the target genes of Wnt/β-catenin signalling and found that the expression levels of *CTNNB1*, *ATF3*, *CCND2* and *MYC* were significantly higher in the ED FB1 subcluster (Fig. 3d), and more JUN and ATF3 positive cells were also detected in ED CC tissues (Supplementary Fig. 10). This evidence suggested that WNT pathway activation may drive the phenotypic transformation between normal and perturbed state.

To investigate this hypothesis, we cultured FBs in vitro with a WNT signalling pathway inhibitor (ICG-001) or activator (SKL2001). The change in β-catenin protein level and its translocation demonstrated the treatment efficiency for WNT signalling pathway (Supplementary Fig. 11b). The cell morphology in both ICG-001-treated and SKL2001-treated groups changed markedly. In the SKL2001-treated group, cells became bigger and a myo-texture appeared, and decreased cell refraction indicated that the FBs were getting thinner (Fig. 3e). The ICG-001-treated group showed an opposite change; the cell area became smaller, the refraction increased, and the internal texture became invisible (Fig. 3e). In addition, a low dose (10 μM) of SKL2001 improved FB proliferation significantly, while ICG-001 treatment severely inhibited FB proliferation (Fig. 3f). To further explore the FB inner changes after activating or inhibiting WNT signalling, we performed transcriptome sequencing of FBs treated with SKL2001 or ICG-001 in vitro. We identified 1711 upregulated and 255 downregulated genes in SKL2001-treated FBs when setting the thresholds of expression fold change >2 and *P*-value <0.05 (Supplementary Data 10). In addition, the GO terms "muscle contraction" and "steroid biosynthetic process" were found positively and negatively enriched, respectively (Fig. 3g). In ICG-001-treated FBs, 1064 upregulated and 803 downregulated genes were identified, and the "collagen trimer" term was significantly inactive (Supplementary Fig. 11c). The trend of muscle contraction and collagen-related gene upregulation was more significant when comparing the SKL2001 and ICG-001 groups (Supplementary Fig. 11d). It is worthy to note that the upregulated genes in pathway "muscle contraction" not only contained the executor of muscle contraction such

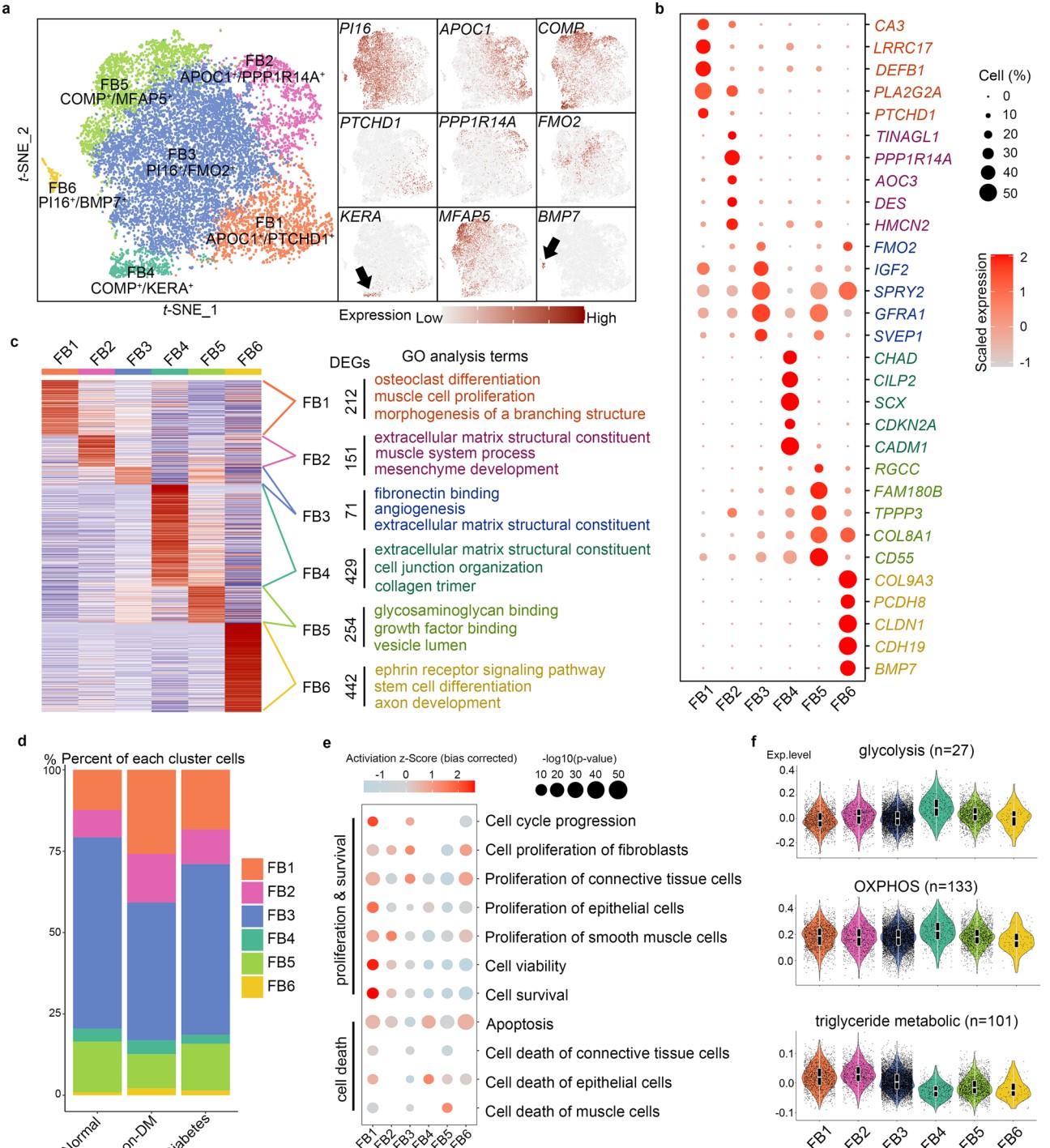

**Fig. 2 | Transcription characteristics and heterogeneity among the six FB subclusters. a** tSNE plots of the six fibroblast (FB) subclusters. Cells are coloured according to their types (left panel) or expression level of the marker gene (right panel). **b** Bubble diagram of the top five DEGs in each FB subcluster. A gradient of grey to red indicates low to high expression levels. The size of bubbles indicates the percentage of cells positive for DEGs in each subcluster. **c** Heatmap showing the DEGs in each FB subcluster (left panel) based on normal samples. The GO terms (biological process) associated with their DEGs are listed in the right panel. The count of DEGs for each FB subcluster are listed in the middle panel. **d** Bar plot showing the cell count proportion of each FB subcluster in normal and ED CC. **e** Cell function (proliferation and death) predicted by IPA analysis according to the DEGs of each FB subcluster shown as a bubble diagram. A gradient of light blue to red indicates inhibition to activation of the term. The size of the bubble indicates the *P*-value. **f** Violin plot combined with box plot showing the expression score of the three major energy metabolism pathway genes in each FB subcluster. Box plots indicate median (middle line), 25th, 75th percentile (box) and 5th and 95th percentile (whiskers) as well as outliers (single points).

as *ACTA2* and *CACNA1H,* but also contained some regulatory signal molecules such as *EDN1.* Neither the constriction of the supplying artery muscle nor of the CC SMC is conducive to penile erection, so this result may represent a pathological state of CCFB.

Increased expression of extracellular matrix proteins and increased proliferation of fibroblasts can lead to disruption of micro-environmental homeostasis and tissue fibrosis. To further explore the effect of WNT activation and inhibition on the whole cavernous tissue

microenvironment, we used SKL2001 and ICG-001 in the CC tissue culture system. The ratio of smooth muscle to collagen was used as a measure of CC tissue fibrosis. As expected, ICG-001 treatment significantly delayed the CC's autonomic fibrosis process in vitro, while SKL2001 showed an opposite effect (Supplementary Fig. 11e, f). These results suggest that WNT signalling could be a drug target for FB phenotype transformation-induced CC structure damage.

## Heterogeneity of muscle contraction regulation and different spatial location in the SMC cluster

The degree of erection depending upon the balance between the inflow and outflow of blood to the penis, and controlled relaxation of the cavernosal smooth muscle is the key process. Because PC cluster also showed general myocyte features, it was combined with SMC cluster and analysed in this part. After re-clustering and

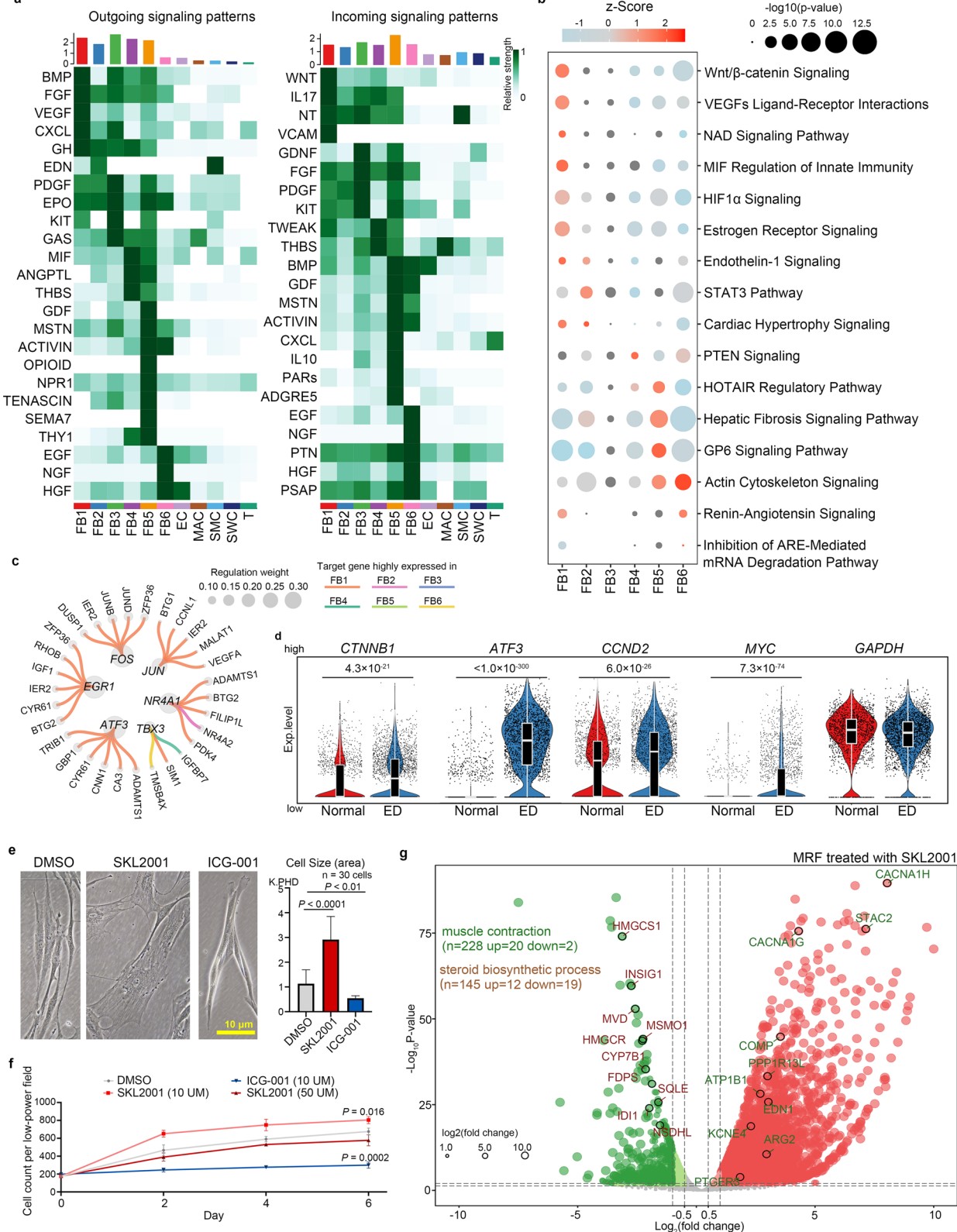

**Fig. 3 | Regulatory signalling of FB phenotypic transformation. a** Heatmap of the CellChat signalling in each FB subcluster. The left panel shows the outgoing signalling patterns (expression weight value of signalling molecules) and the right panel shows the incoming signalling patterns (expression weight value of signalling receptors). A gradient of white to dark green indicates low to high expression weight value in the heatmap. **b** Bubble diagram of the top activated IPA pathway in each FB subcluster. A gradient of light blue to red indicates inhibition to activation of the term. The size of the bubble indicates the *P*-value from high to low. **c** The top candidate master regulators and their target genes for the FB1 subcluster. The bubble size indicates the weight from high to low. **d** Violin plot combined with box plot showing the expression level of the genes downstream of the WNT pathway between normal male and ED patients. The housekeeping gene *GAPDH* is also listed as a reference. Box plots indicate median (middle line), 25th, 75th percentile (box) and 5th and 95th percentile (whiskers) as well as outliers (single points). The statistical analysis was made by Wilcoxon (Mann–Whitney) rank-sum test; two-tailed; the confidence interval is 95%. **e** The morphology of cultured FBs after SKL2001 ($P < 0.0001$) or ICG-001 treatment ($P = 0.0015$). Data are shown as mean ± SD. The scale bar represents 10 μm. The statistical analysis was made by ANOVA with Tukey's multiple comparisons test; two-tailed; the confidence interval is 95%. **f** The statistical analysis of cell count in the group treated with SKL2001 ($P = 0.016$) or ICG-001 ($P = 0.0002$) for 7 days, data were shown as mean ± SD, $n = 3$ independent experiments. The statistical analysis was made by ANOVA with Tukey's multiple comparisons test; two-tailed; the confidence interval is 95%. **g** Volcano plot showing the DEGs between FBs treated with SKL2001 and negative controls by RNA sequencing.

dimensionality reduction processes, we found that the SMC could be further divided into four subclusters (Fig. 4a). Even though all four subclusters showed a high level of ACTA2 and MYH11, each subcluster showed a distinctive transcription pattern of some specific actin and myosin proteins (Fig. 4a). In the four SMC subclusters at normal baseline, 531, 383, 345 and 711 DEGs were identified, respectively (Supplementary Data 11). GO analysis showed that the GO term "vasculogenesis" and "artery development" was enriched in the SMC1 and SMC3 subclusters, suggesting their role in angiogenesis (Fig. 4b).

To further determine the spatial location information for each SMC subcluster, we chose desmin (DES), a SMC2-specific protein, to mark this cluster with IHC in the corpus cavernosum tissue. Interestingly, DES+ cells were detected in the nearby septum pectiniforme and cavernosal trabecular regions but not in the vascular region, indicating that DES⁺ SMCs were the cavernosal trabecular SMCs (Fig. 4c). Furthermore, *THY1* was expressed highly in SMC3 and part of FB, and THY1+ cells were detected around small vessels where fits the location of pericytes (Supplementary Fig. 12a). In addition, we found that LUM, a collagen fibril binding protein that is distributed in interstitial collagenous matrices throughout the body, was highly expressed in all FBs and in SMC4 (Supplementary Fig. 12b). LUM+ cells were widely distributed in the cavernosal trabecular region but not in the small artery (Supplementary Fig. 12b). Combined the above results and their transcript characteristic, we inferred that SMC1 is the cluster of vascular SMCs (VSMC), SMC2 are the cavernosal trabecular SMCs (CCSMC), SMC3 are pericytes (PC) and SMC4 are a type of myofibroblast (MFB).

Continuous contraction of the penis smooth muscle is an important cause of ED, so we then compared the activation and inhibition of various signalling pathways affecting smooth muscle contraction and relaxation in normal male and ED patients. For the contraction pathways, the alpha adrenoceptor showed significant heterogeneity among the four SMC subclusters. *ADRA2A* was only detected in VSMC and FB, and its expression level was significantly higher in DMED patients than in normal within the MFB cluster. However, *ADRA2C*, another subtype of alpha adrenoceptor, was highly expressed in CCSMC, and its expression level was also upregulated in DMED patients but not in non-DM patients (Fig. 4d). *MYLPF* (myosin light chain, phosphorylatable, fast skeletal muscle) and *MYLK* (myosin light chain kinase) are involved in maintaining continuous smooth muscle contraction, and both were highly expressed in CCSMC, especially in ED patients (Fig. 4d). Other muscle contraction and maintenance pathways, such as the Rho/ROCK pathway, thromboxane signalling, and endothelin signalling, also showed cell type heterogeneity within the SMC cluster. For example, *ROCK1* was highly expressed in VSMC, *TBXA2R* was highly expressed in CCSMC, and *EDNRA* was highly expressed in PC (Supplementary Fig. 12c–e). It is noteworthy that even though *EDN1* is a potent vasoconstrictor and has been reported to promote contraction of smooth muscle in the CC, we found that its receptor *EDNRA*

was downregulated in both non-DM and DMED patients (Supplementary Fig. 12e).

In addition to the classical signalling pathways associated with erections, we used IPA analysis to comprehensively describe the different signalling in CCSMC between normal male and ED patients according to the DEGs (Supplementary Data 12). The intersection of pathways that changed in both non-DM and DMED patients was identified, and "TGF-β signalling", "ECM–receptor interaction" and "protein targeting" were activated, while "FoxO signalling", "mineral absorption" and the "Hippo signalling pathway" were significantly inhibited in ED patients (Supplementary Fig. 13a). The Hippo signalling pathway is involved in the pathophysiology of various cardiovascular diseases[22]. *YAP1* (Yes-associated protein), the main transcriptional complex of the Hippo pathway, was detected in VSMC, CCSMC, and MFB (Supplementary Fig. 13b). Furthermore, we found that *CTGF*, a downstream target gene of the YAP1 complex and a potent skeletal muscle fibrosis promotor[23], was significantly upregulated in all four SMC subclusters of ED patients (Supplementary Fig. 13b). These results were further confirmed by IHC staining of cavernous tissue. In normal male samples, CTGF+ cells were only detected in the interstitial area between smooth muscle bundles, but almost all SMCs in the CC of ED patients were CTGF+ (Supplementary Fig. 13c). The expression of YAP localized in the CCSMC nucleus of ED patients was significantly elevated (and the volume of these nuclei increased), while p-YAP expression levels were decreased (Supplementary Fig. 13d).

## Heterogeneity of biological characteristics and spatial location in the EC cluster

ECs play an important role in smooth muscle contraction regulation and vascular barrier structure, and injury to ECs is a major cause of ED, especially DMED[24]. After re-clustering and tSNE dimensionality reduction, the EC cluster under normal state was further divided into four subclusters (Fig. 5a). A total of 220, 151, 699 and 662 DEGs were identified in each EC subcluster (Fig. 5b and Supplementary Data 13). And the DEGs between normal and ED patients in each EC subcluster were also identified (Supplementary Fig. 15). Based on these DEGs, we performed GO analysis and found that the terms "migration" and "vascular development", which belong to the basic functions of ECs in the vascular system, were enriched in EC1 (Fig. 5b). However, in the EC4 subcluster, GO terms such as "extracellular matrix organisation", "response to TGFβ", and "mesenchyme development" were enriched. Furthermore, a large number of ECM- and myo-related genes, such as *COL1A1*, *COL5A2*, *LUM*, *ACTA2*, *ACTG2* and *DES*, were upregulated in EC4, indicating that EC4 underwent endothelial–mesenchymal transition (Fig. 5b and Supplementary Data 13). Finally, GO terms such as "response to cytokine", "regulation of cell death" and "cell adhesion molecules" were enriched in EC2, and some cytokine receptors and adhesion molecules, such as *ACKR1*, *LIFR*, *LTBR*, *IL1R1*, *CD34*, *SELP*, *SELE* and 12 MHC-II genes, were highly or specifically expressed in EC2 (Fig. 5b and Supplementary Fig. 16a). In summary, EC2 and EC4 were

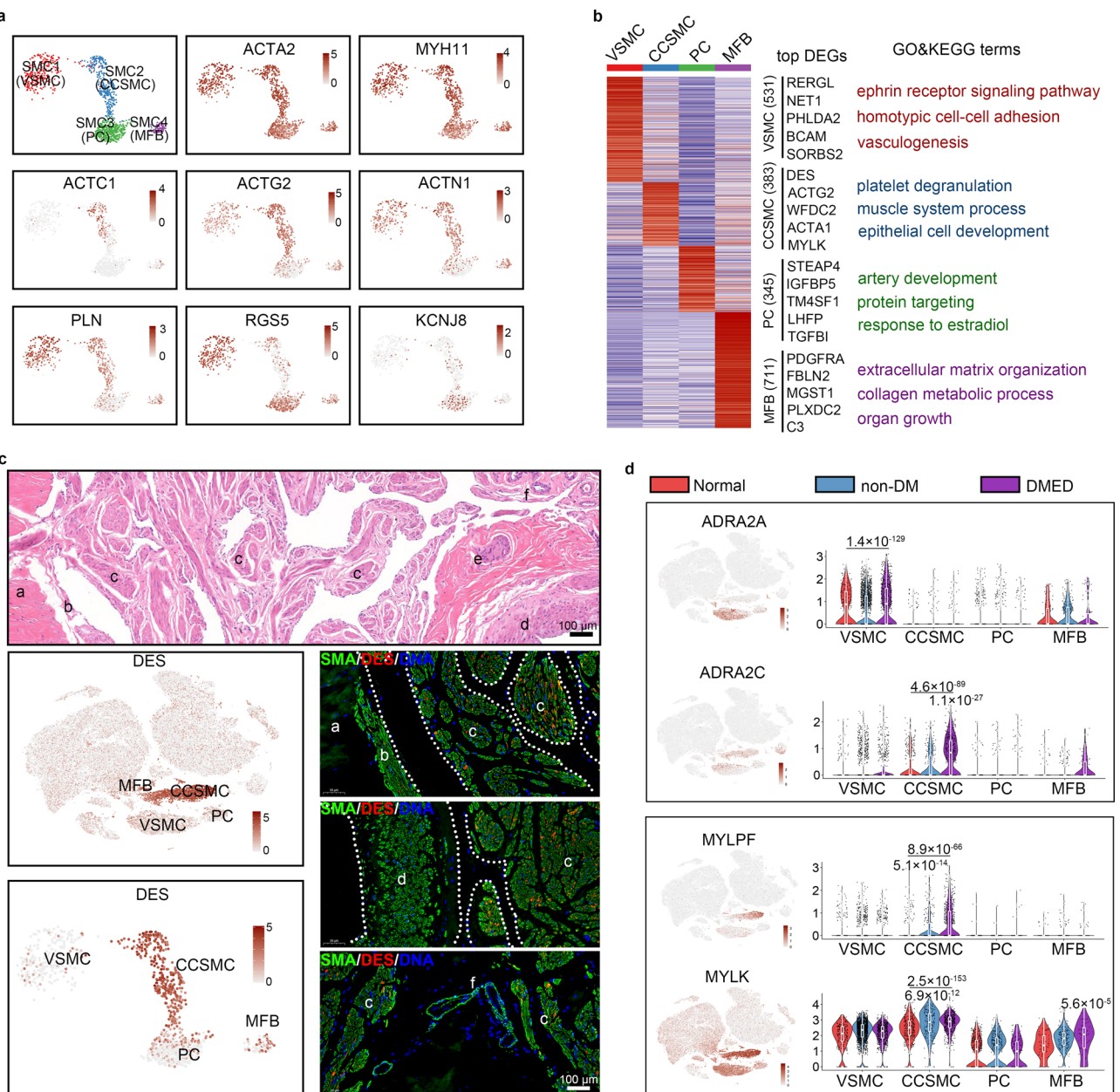

**Fig. 4 | The heterogeneity within the SMC cluster. a** tSNE plots of the four SMC subclusters. The cells are coloured according to their subcluster or the expression level of classic actin and myosin genes. VSMC: vascular smooth muscle cells; CCSMC: corpus cavernosal trabecular smooth muscle cells; PC: pericyte; MFB: myofibroblast. **b** The DEGs in each SMC subcluster (left panel) based on normal samples are shown as a heatmap, and the GO terms (biological process) associated with their DEGs are listed in the right panel. The count of the DEGs for each SMC subcluster is listed in the middle panel. **c** CCSMC and VSMC could be distinguished according to DES protein expression. H&E staining showing the main region of the CC (upper panel). tSNE plots showing that DES is specifically expressed in the CCSMC subcluster (left panel). Immunofluorescence co-staining of DES (red) and SMA (green) in CC paraffin sections. The scale bar represents

100 μm. a, septum pectiniforme; b, region near septum pectiniforme; c, cavernosal trabecular region (sinusoid); d, cavernous artery; e, nerve bundles; f, small vessels. **d** The heterogenous expression pattern of smooth muscle contraction-related genes in the four SMC subclusters. *P*-values with a horizontal line on the violins represent statistical differences between this subcluster and the other subclusters within the SMC cluster. The statistical analysis was made by ANOVA with Tukey's multiple comparisons test; two-tailed; the confidence interval is 95%. *P*-values without a line on the violins represent a significant difference between this disease type and the two other types within one subcluster. Box plots indicate median (middle line), 25th, 75th percentile (box) and 5th and 95th percentile (whiskers) as well as outliers (single points).

more like ECs in pathological states, and the increased proportion of these two subclusters in ED patients further suggests their relationship with cavernosal lesions.

Then, we investigated the different spatial localisation of each EC subcluster. The IHC staining result showed that *KIT*, an EC1 marker gene, was specifically expressed in the cavernosal trabecular EC (CCEC), but not in vascular EC (VEC) (Fig. 5d). However, it should be noted that the mast cells (mast tryptase-positive cells) distributed in

the cavernosal trabecular region were also KIT-positive cells and could be distinguished using general endothelial markers such as VWF and CD31. On the contrary, GJA5 was a EC3-specific marker, and GJA5+ ECs were only detected in arteries and their branches, indicating that EC3 are VEC (Fig. 5e). The human corpus cavernosum is supplied directly by arterioles and lacks capillaries. We further investigated the expression of different vascular endothelial marker genes in the four EC subclusters based on previous reports and

found that the EC3 subcluster indeed showed transcription characteristic of artery ECs. However, all these marker genes lacked characteristic transcriptional distribution in the EC1 subcluster, suggesting that the CCEC were a unique endothelial subtype (Supplementary Fig. 16b)[25,26]. In biopsies of ED corpus cavernosum, we found that the high expression of IGF1 may be related to endothelial injury because the cavernosum tissue of ED patients contained larger

and more IGF1-positive regions than that of normal males. In the high IGF1 expression region, VWF+ ECs disappeared and the FBs were exposed, but this phenomenon was not detected in the low IGF1 region. In addition, IGF1 and VWF double-positive cells were mainly found in the adjacent regions with endothelial deficiency (Fig. 5f, g). IGF1 and its receptor signalling pathway-related genes were highly expressed in the EC4 subcluster (Fig. 5h). IGF1 inhibits EC apoptosis

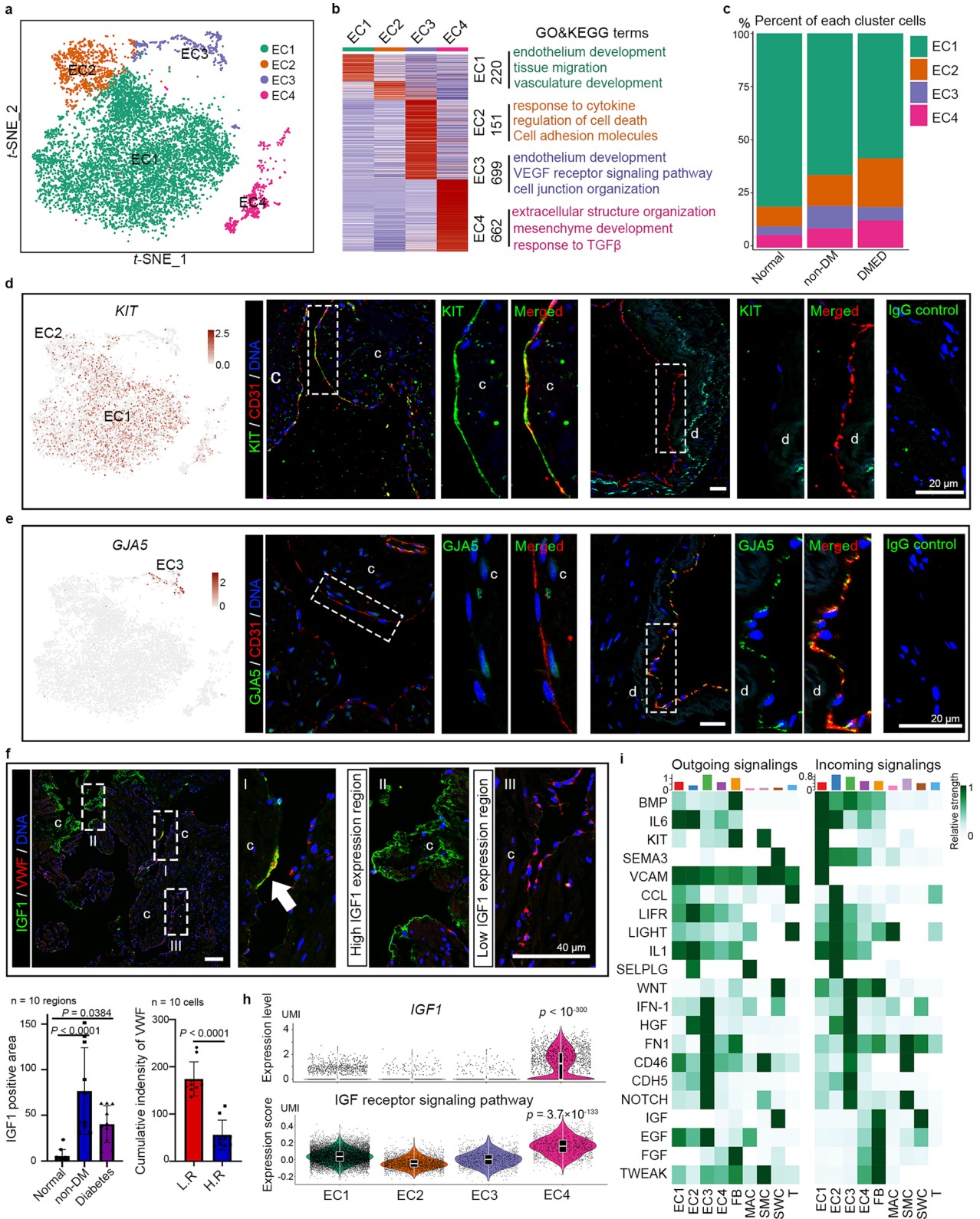

**Fig. 5 | The heterogeneity within the EC cluster. a** tSNE plots of the four EC subclusters. **b** The DEGs in each EC subcluster (left panel) are shown as a heatmap, and the GO terms (biological process) associated with their DEGs are listed in the right panel. **c** Bar plot showing the proportion of each FB subcluster in each disease group. **d, e** tSNE plots showing the expression pattern of **d** *KIT* and **e** *GJA5* in the EC cluster. Immunofluorescence co-staining of **d** CD31 (red)/KIT (green) and **e** CD31 (red)/GJA5 (green) in corpus cavernosum paraffin sections. The right panel is an enlargement of the dotted box in the middle panel. The scale bar represents 20 μm. **f** Immunofluorescence co-staining (upper panel) of VWF (red) and IGF1 (green) in CC paraffin sections. The right panels are an enlargement of the dotted box in the left panel. The arrow marks VWF/IGF1 double-positive cells (EC4). The

scale bar represents 40 μm. c, sinusoid (cavernosal trabecular); d, cavernous artery. **g** The size of the IGF1-positive region and the immunofluorescence intensity of VWF in the IGF1-negative region (L.R.) or in the IGF1-highly expressed region (H.R.). Data are shown as mean ± SD. $n = 10$ different regions or cells. The statistical analysis was made by ANOVA with Tukey's multiple comparisons test; two-tailed; the confidence interval is 95%. **h** The transcript level of *IGF1* signalling genes shown as a violin plot combined with a box plot. Box plots indicate median (middle line), 25th, 75th percentile (box) and 5th and 95th percentile (whiskers) as well as outliers (single points). The statistical analysis was made by Wilcoxon (Mann–Whitney) rank-sum test; two-tailed; the confidence interval is 95%. **i** Heatmap of the CellChat signalling in each EC subcluster.

and promotes vascular injury repair in various tissues and organs[27–29], indicating that an increase in the local IGF1 concentration and the emergence of EC4 were feedback regulations in response to endothelial injury.

To investigate the upstream regulatory signalling in each EC subcluster, we further described the signalling network between the four EC subclusters and the other corpus cavernosum cells. The CellChat result showed that "BMP", "IL6" and "KIT" signalling were enriched in the EC1 subcluster. Immune-related signalling pathways, such as "CCL", "LIFR" and "LIGHT (TNFSF14-TNFRSF14/LTBR)" signalling, were activated in EC2. In addition, "WNT", "NOTCH" and "IFN-1" signalling and "IGF", "EGF" and "FGF" signalling were dominant in EC3 and EC4, respectively (Fig. 5i).

### Mitochondrial dysfunction and apoptosis are major pathological features in the cavernosal trabecular endothelium of ED patients

Similar to other vascular systems, the different EC subclusters showed an obvious heterogeneous susceptibility for injury and aging[30]. The cell function results of the IPA analysis showed that the EC3 and EC4 subclusters had a more active cell proliferation and migration ability, while the IPA term "cell death" was activated in the EC2 subcluster, suggesting that the CCEC had a higher susceptibility to injury (Supplementary Fig. 16c). This hypothesis was further validated by IHC staining. We found that the cavernosal endothelium of both normal and ED patients had good endothelial integrity and endothelial marker expression in the arteries and their branches; however, decreased expression of endothelial markers and many endothelial exfoliation regions were detected in ED samples, especially in DMED patients (Supplementary Fig. 16d).

To get a more intuitive impression, scanning electron microscopy (SEM) was used to detect endothelial changes in normal and ED CC. In normal tissue, the endothelial surface was smooth and the EC were arranged in a neatly manner with narrow gaps. However, in both non-DM and DMED tissue, the cavernosal trabecular endothelium was obviously wrinkled, and particles with diameters of 50–100 nm were densely adhered to its surface. Some regions of the endothelium were detached with the collagen fibres inside exposed. In DMED tissue, the ECs became obviously swollen, forming a papillary bulge on the surface (Fig. 6a). In addition, the results of γH2AX immunofluorescence staining showed that the ECs of ED cavernosal trabeculae underwent a significant increased apoptosis (Fig. 6b).

Oxidative phosphorylation dysfunction, which induces oxidative stress and decreases metabolism, is an important cause of endothelial injury in ED CC[31]. In the IPA analysis, we found that *NCOA7*, *SFPQ*, *MCL1*, *TXN* and *SOD2*, which belong to the "cell death in response to oxidative stress" pathway, were significantly upregulated in ED EC, and no genes belonging to this pathway were downregulated (Fig. 6c). Furthermore, the "oxidative phosphorylation" pathway was significantly inhibited in the EC3 subcluster of ED patients (Fig. 6d). To explore the cause of this energy metabolism disorder, a transmission electron microscope (TEM) was used to detect the shape and structural changes of mitochondria in the cavernosal trabecular ECs. In ED

ECs, the mitochondria were markedly swollen, characterised by increased mitochondrial size and decreased density under TEM. In addition, the number of mitochondria per EC decreased in the non-DM ED patients (Fig. 6e). Integrated analysis of genes in the glycolysis and oxidative phosphorylation pathways showed that the glycolysis gene set was significantly upregulated in DMED ECs, while the oxidative phosphorylation gene set was significantly downregulated in both non-DM and DMED patients (Fig. 6f and Supplementary Fig. 16e). In summary, cell apoptosis and mitochondrial dysfunction are the main pathological features of endothelial injury in ED CC.

### YAP signalling is involved in endothelial injury of DMED CC

In addition to EC injury, fibrosis is another important pathological feature in DMED CC. EC has been proved as one of major sources to the emergence of fibroblasts[32]. We found that many fibrogenic signals (Supplementary Data 8), such as "integrin signalling", "TGF-β signalling" and "renin–angiotensin signalling" in CCEC of ED patients were activated[33–35], while negative modulating pathways of fibrosis, such as "HIPPO signalling" and "PPAR signalling", were inhibited (Fig. 6d)[36,37]. Furthermore, the signalling network analysis based on the DEGs of the EC4 subcluster also indicated that "TGFB3" and "AGT (angiotensinogen)", extracellular signalling factors belonging to "TGF-β signalling" and "renin–angiotensin signalling", were upregulated. In the intracellular region, their downstream transcription factors such as "TEADs" and "SMAD3" were also activated (Supplementary Fig. 17a). YAP is the most important downstream nuclear effector of the Hippo signalling pathway, and can bind to TEADs or SMAD3 to regulate downstream gene expression, and then induce fibrosis and endothelial dysfunction[38]. From the IHC staining, we found that the expression level of nuclear-localised YAP protein was significantly increased and phosphorylated YAP was decreased in ED EC, suggesting that the Hippo pathway is indeed inhibited (Fig. 6g and Supplementary Fig. 17b).

Since abnormal inactivation of the Hippo pathway is associated with endothelial injury in ED corpus cavernosum, we further verified the effects of Hippo activation or inhibition on ECs in vitro. After isolation and purification, cavernosal trabecular ECs were treated with Peptide 17 (a YAP-TEAD inhibitor) or XMU-MP-1 (a MST1/2 inhibitor that can activate downstream YAP protein). We found that neither of the two drugs alone obviously altered the morphology and biomarker expression (Supplementary Fig. 8c). We therefore further tested their effects in the presence of TGF-β, the most common inducer of fibrosis. We found that ECs treated with TGF-β became larger in size, became dispersed rather than close together, and the expression of endothelial markers decreased. This process was inhibited and promoted by Peptide 17 and XMU-MP-1 treatment, respectively (Fig. 6h). In addition, TGF-β and XMU-MP-1 treatment also increased permeability of CCEC (Supplementary Fig. 5e). This result suggested that a YAP inhibitor could alleviate the injury to ECs induced by TGF-β. Next, we used CC tissue culture experiment which theoretically contains all the cellular components and signalling networks in the microenvironment, to the maximumly extent possibility of simulating the effects of these factors on CCEC and CC tissue in vivo. We found that

TGF-β, Angiotensin II and XMU-MP-1 significantly promoted the progression of tissue fibrosis, while Peptide 17 showed the opposite effect. Two other YAP inhibitors, Verteporfin and TED-347, could also alleviate fibrosis of the CC tissue; however, both their short-term and long-term efficacies were inferior to that of Peptide 17 (Supplementary Fig. 17d).

## Discussion

The CC has been extensively studied in model organisms, such as mice and rats. However, our knowledge of the cellular heterogeneity and communication in the human CC at the molecular level and at a single-cell resolution are still limited. Here, we used scRNA-seq to produce a single-cell transcriptomic atlas of CC tissue from normal males and ED patients. These data provide insights into the cellular composition, biological features, and regulatory signalling networks of the CC, to give a better understanding of the human CC microenvironment and its dynamic changes in ED patients.

In this study, we identified seven major cell subsets in eight human CC samples. Most of these cell subsets have been reported to play crucial roles in physiology and diseases[9,39,40]. Furthermore, we found the FB cluster, the dominant cavernosal cell subset, as a major signalling output source in CC microenvironment rather than

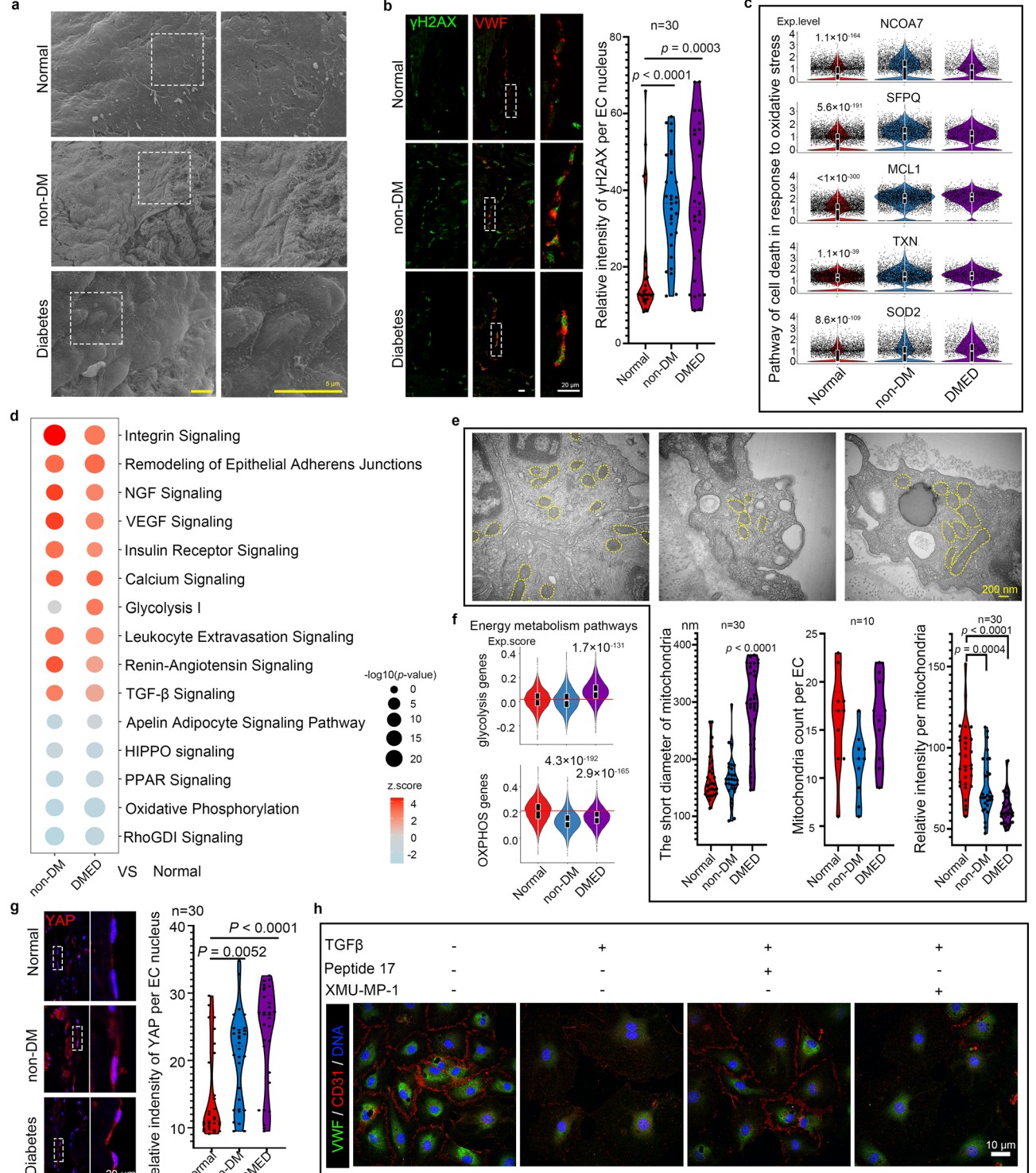

**Fig. 6 | Pathological features of endothelial injury in ED CC. a** Scanning electron microscope image of normal and ED cavernosal trabecular endothelium. The scale bar represents 5 μm. **b** Immunofluorescence co-staining of VWF (red) and γH2AX (green) in CC paraffin sections. The statistical analysis of relative fluorescence intensity per EC nucleus ($n = 30$ cells in 5 different regions of each group) was made by ANOVA with Tukey's multiple comparisons test; two-tailed; the confidence interval is 95%. The scale bar represents 20 μm. **c** Violin plot combined with box plot showing the expression level in each EC subcluster. The statistical analysis was made by Wilcoxon (Mann–Whitney) rank-sum test; two-tailed; the confidence interval is 95%. **d** Bubble diagram of the top activated and inhibited IPA pathways based on the DEGs of ED EC compared with normal EC. Statistical analysis was based on Fisher's exact test; two-tailed; the confidence interval is 95%. **e** Transmission electron microscope image of normal and ED cavernosal trabecular endothelium. The bottom panel shows the statistical analysis of the short

diameter per mitochondrion ($n = 30$ mitochondria), mitochondria count per EC ($n = 10$ cells), and relative intensity per mitochondrion ($n = 30$ mitochondria) made by ANOVA with Tukey's multiple comparisons test; two-tailed; the confidence interval is 95%. The scale bar represents 200 nm. **f** Violin plot combined with box plot showing the expression level of the genes that are associated with energy metabolism. Statistical analysis between the normal EC and the other two groups was made by two-tailed Wilcoxon rank-sum test in R. **g** Immunofluorescence staining of YAP (red) in CC paraffin sections. The statistical analysis of relative fluorescence intensity per EC nucleus in normal and ED CC was made by ANOVA with Tukey's multiple comparisons test based on 30 ECs in 5 different fields; two-tailed; the confidence interval is 95%. The scale bar represents 20 μm. **h** Immunofluorescence co-staining of CD31 (red) and VWF (green) in cavernosal trabecular ECs treated with TGF-β, Peptide 17, and/or XMU-MP-1. The scale bar represents 10 μm.

simply forming the scaffold for CC tissue by expressing the extracellular matrix protein. They produce VEGF, BMP and IGF signalling to regulate ECs, and produce NGF signalling to regulate neuron and Schwann cells. The FBs are also involved in the response to injury; for example, they secrete large amounts of IGF1 protein in regions with endothelial deficiency to promote injury repair. This evidence suggests that the FBs have more functions than simply extracellular matrix formation. In addition, the spatial heterogeneity within the FBs suggests that they may be derived from different precursor cells and play different roles in the formation of the cavernosum structure. To clarify this hypothesis, it would be necessary to collect samples at different ages (from the foetus to post-puberty) for single-cell sequencing. Furthermore, the change of proportion of each FB subcluster in ED corpus cavernosum suggests that abnormal phenotypic transformation within the FB cluster may destroy the homeostasis of the microenvironment. The standardized naming of FB subcluster across tissues referenced Buechler's study, however, considering that CC was a specific type of vascular structure, additional dataset of heart, muscle, skin (contain small vessels) and kidney were compared in this study, and a cluster (APOC1+ FB) enriched in CC, heart, and muscle was identified. The apparent increase in APOC1+/PTCHD1+ FB (FB1) cluster in ED patients indicated their relevance to the CC tissue injury or repair. This cluster also expressed high level of muscle-related genes; it may be related to the CC sinusoids dilatation limitation that similar to restrictive cardiomyopathy[41]. However, the similarity between CC and heart highlights the current challenges of defining CC FB subsets by only transcriptomic technologies, for example, the FB4 cluster expressed a high level of *SCX*, while Scx+ cells have been identified as important in progenitors for cardiac valve morphogenesis[42]. Further spatial transcriptome combined with multiomics analysis would solve many limitations left in this part. In addition, we identified some signalling pathways that may play an important role in this process. For example, WNT/β-catenin signalling promotes the fibroblast-to-myofibroblast transition during cardiac, pulmonary, and renal fibrosis[43–45], and it was activated in the FB1 subcluster. In this study, FB treated with WNT signalling inhibitor did not showed a classical myofibroblast phenotype: although most muscle-contraction- and ECM-related proteins were upregulated, some typical myofibroblast upregulated genes such as COL1A1 and LOX were downregulated. However, the delaying effect of ICG-001 on the fibrosis of CC tissue ex vivo still suggested that the WNT pathway may serve as a potential drug target for inhibiting cavernous fibrosis in ED patients.

The contraction and relaxation of smooth muscle is the key event in the erection process[46,47]. Unlike rats, human smooth muscle bundles are larger and occupy most of the space in the cavernosal trabecular region, rather than being banded in the subendothelial region[40]. Because of the absence of the penile bone, the human cavernosal smooth muscles are subjected to tremendous pressure compared with those in other tissues, usually higher than systolic

blood pressure in the rigidity phase of an erection, to maintain erection and complete insertion[2]. We observed a significant difference in desmin protein expression in the cavernosal trabecular region. Desmin filaments form the intermediate filament system of muscle cells where they play important roles in maintaining mechanical integrity and elasticity. Desmin mutations lead to entangled filaments and impaired tensile properties[48,49]. These results suggest that the CCSMC is a special kind of smooth muscle rich in desmin intermediate filaments, which may be the structural basis of its ability to withstand the drastic changes in length and pressure during the cavernous sinus filling process. In addition, there were significant differences in the distribution of some classic signalling receptors that regulate SMC contraction and relaxation among SMC subclusters, such as adrenergic receptors. This heterogeneity may help the development of therapeutic drugs that only target cavernosal trabecular smooth muscle, to reduce the multiple target effects of the current drugs that are commonly used, such as headaches, gastrointestinal complaints, and hypotension[50,51].

The number of ECs was comparable to that of FBs, accounting for about 40% of the total cavernosal cells. Even though the heterogeneity of ECs among different tissues or within one organ has been reported[25,52,53], whether human cavernosal ECs are also heterogeneous has remained unclear because of limited access to adult human tissue. In this study, we identified four EC subclusters, by comparing the sequencing data from multi-tissue-derived ECs[25], we found that the expression profile of CCEC was significantly different from that of arteries or veins, especially regarding the lack of GAP junction protein GJA5, suggesting a unique mechanism of signal transduction and endothelial junctions. Endothelial injury is the most significant pathological change of corpus cavernosum disorder, which may occur in the early stage of ED[24]. We found that the CCEC were more susceptible than the adjacent vascular ECs (EC3), providing a histological explanation for the clinical observation that ED shares risk factors with cardiovascular disease and could be a harbinger of cardiovascular events. However, the molecular mechanism underlying the differing susceptibilities still requires further exploration. The endothelial fibroblasts (EC4) only existed in the region of endothelial shedding and high IGF levels, indicating that they may be involved in endothelium repair under the regulation of paracrine IGF signalling.

In ED samples, we observed a regulatory pattern of activated TGF-β signalling and inhibited Hippo signalling in both the SMC and EC clusters. The crosstalk between Hippo/YAP and TGF-β/SMAD3 signalling extensively affects the fibrosis process of various tissues and organs[36,38,54]. We found that Peptide 17 or XMU-MP-1 treatment could relieve or accelerate the endothelial injury and cavernosal fibrosis induced by TGF-β treatment in vitro, indicating that Hippo/YAP could be another candidate drug target for organic lesions of the corpus cavernosum.

Our study reveals the human corpus cavernosum's transcriptome landscape and cellular heterogeneity and offers insights

into the cellular foundation of normal erection and ED knowledge. Supported by verification at the protein level and cell function experiments, we also highlight some regulatory signalling pathways and therapeutic targets.

## Methods

### Experimental models and subject details

The experiments performed in this study were approved by the Ethics Committee of Shanghai General Hospital (License No. 2021SQ259). The Office of Academic Research and the Ethics Committee of Shanghai General Hospital have authorised this study and confirmed the study design and conduct complied with all relevant regulations regarding the use of human study participants and was conducted in accordance with the criteria set by the Declaration of Helsinki. For CC samples, all donors signed their consent after being fully informed of the goal and characteristics of our study. We have obtained all donors' consent to publish information that identifies individuals including sex, age, BMI and the hospital name. Three normal tissue samples were obtained from the tumour margin of penile carcinoma resection, all patients reported good stimulated erections and early morning erections. The five ED CC tissues were obtained from biopsy samples in artificial cavernous body implantation, all the ED patients were diagnosed with organic ED rather than psychological ED by the nocturnal penile tumescence (NPT) and intracavernosal injection (ICI) tests. The DMED patients had an at least 10-year history of type 1 diabetes and had good blood glucose control before surgery. The detailed clinical information of these patients could be found in Supplementary Data 1.

### Histological examination

Fresh CC tissues obtained from surgery were washed with PBS twice to remove blood and immediately fixed in 4% paraformaldehyde for 12 h at 4 °C, then embedded in paraffin and sectioned. Before staining, the tissue sections were dewaxed in xylene, rehydrated using a gradient series of ethanol solutions, and washed in distilled water. For Masson staining, the sections were stained in turn with haematoxylin, aniline blue, and ponceau. For Sirius red staining, the sections were stained with Sirius red and haematoxylin. Then, sections were dehydrated using increasing concentrations of ethanol and xylene, and allowed to dry before applying neutral resin to the coverslips. Images were captured with a Nikon Eclipse Ti-S fluorescence microscope (Nikon). The images were measured and analysed with Adobe Photoshop CC 2018 software. In brief, the muscle/collagen ratio was represented by the area and identity of the red region (colour code = #0068b7; colour range = 160%) and the blue region (colour code = #b26c91; colour range = 160%), respectively.

### Immunohistochemical staining and fluorescence intensity measure

Sections were performed with rehydration, antigen retrieval, and blocking, and then incubated with appropriate primary antibodies (Supplementary Data 15) at 4 °C overnight. Sections were further incubated with secondary antibody for 2 h at room temperature. Nuclei were labelled with Hoechst33342 by incubating tissue sections for 15 min[55]. Images were captured with an OLYMPUS IX83 confocal microscope.

Fluorescence intensity measurements were processed with Adobe Photoshop CC 2018 software. For the fluorescence intensity of β-catenin and YAP, the nuclear regions were selected, for phosphorylated YAP, CD31 and VWF, the whole cell regions were selected, and then the single staining of the target protein was converted to greyscale. After that, the fluorescence intensity of the chosen regions was recorded.

### Isolation of single CC cells

Fresh CC tissue samples of about 1 × 1-cm size were first immersed in 15 ml PBS with vigorous shaking to remove residual blood cells. Then, tissues were cut into pieces of 1 × 1-mm size and enzymatically digested with 2.5 mg/ml collagenase type I (17104-019, Gibco), 4 mg/ml collagenase type IV (17018029, Gibco), 0.1 mg/ml neutral protease, and 2 mg/ml DNase I (#AMPD1, Sigma-Aldrich) at 37 °C for 40 min. Subsequently, the cell suspension was filtered through a 40-mm nylon mesh, and the cells were sorted by MACS with a Dead Cell Removal Kit (130-090-101, Miltenyi Biotec) to remove dead cells. The cells were re-suspended in 0.05% BSA/PBS buffer before 10× Genomics library preparation.

### Single-cell RNA-seq library preparation and sequencing

Single-cell RNA-seq libraries were prepared using the Chromium Single Cell 3′ Library & Gel Bead Kit v3 (PN-1000094, 10× Genomics) according to the manufacturer's instructions. Final libraries were sequenced on an Illumina NovaSeq 6000. The raw sequencing reads were processed by Cell Ranger (v.3.1.0) with the default parameters. The reference genome was GRh38-1.2.0.

### Quality control and sample integration

The gene–cell matrix of each sample was used to create a Seurat object with the Seurat package in R. Cells were further filtered according to the following threshold parameters: the total number of expressed genes, 800–7000; total UMI count, 0–30,000; and proportion of mitochondrial genes expressed, <10%. Batch correction was performed using the *IntegrateData* function in the Seurat package according to the "package manual [https://satijalab.org/seurat/v3.1/pbmc3k_tutorial.html]".

### Cell identification and clustering analysis

The merged Seurat objects were scaled and analysed by principal component analysis (PCA). The first 20 PCs were also used to get clustering and perform t-distributed stochastic neighbour embedding (tSNE) dimensionality reduction. The *FindClusters* function in Seurat package with the resolution parameter set as 0.5 was used to cluster the cells. For further analysis of each cluster, we isolated them and performed the above two steps again to get subcluster information.

For each major cluster, the EC cluster was identified by high expression of *PECAM1* and *VWF*; the FB cluster was identified by expression of *LUM*, *COL1A1* and *PDGFRA*; the PC cluster was identified by high expression of *RGS5* and *KCNJ8*, but low expression of *MYOCD*; the SMC cluster was identified by high expression of *ACTA2* and *MYH11*; the SWC cluster was identified by high expression of *S100B* and *MPZ*; the MAC cluster was identified by high expression of *CD163* and *CD68*; and the T cluster was identified by high expression of *CD3D* and *CD3E*.

### Differentially expressed gene calculation and gene enrichment analysis

The Seurat function *FindAllMarkers* (test.use = wilcox; min.pct = 0.1; logfc.threshold = 0.25) was used to identify differentially expressed genes (DEGs) based on the normalised UMI count. Unless otherwise noted, the DEGs in each selected subcluster were calculated based on comparison between that subcluster and the rest of the dataset. GO analysis was performed using the "*WebGestalt* [website http://www.webgestalt.org]", the Over-Representation Analysis (ORA) or Gene Set Enrichment Analysis (GSEA) was chosen as Method of Interest, and only Biological Process was chosen in Functional Database. The pathway analysis was performed using Ingenuity Pathway Analysis (IPA) software based on the $\log_2$ (FC) and *P*-values of the DEGs. The other gene sets used in this study are listed in Supplementary Data 15.

### Ligand–receptor interaction and transcription factor network construction

The CellChat package was used for ligand–receptor interaction analysis. The cell–gene matrix was divided according to the six major clusters or five major combined with the subclusters of the other major cluster. The "secreted signalling", "ECM–receptor" and "cell–cell contact" paired datasets were chosen to analyse the cell communication.

For the transcription factor (TF) regulation network analysis, a total of 1469 human TFs in AnimalTFDB and a cell–gene matrix were taken as input for the GENIE3 package. In the output regulator–target table, only pairs with weights greater than 0.1 were retained and the first 2–6 TFs were displayed further.

### CC tissue culture experiment

Fresh CC tissue samples of about $1 \times 1$-cm size were first immersed in 15 ml PBS with vigorous shaking to remove residual blood cells. Then, tissues were cut into pieces of $1 \times 1$-cm size and cultured in 2% FBS/DMEM with SKL2001/ICG-001 (for the results of Supplementary Fig. 11f) or ANG II, TGFβ, Peptide 17, Verteporfin, TED-437 and XMU-MP-1 (for the results of Supplementary Fig. 17d). The culture systems were than place in a place on a shaking table in an incubator, 5% $CO_2$, 37 °C. A quarter of the tissue was collected for Masson or H&E staining on the 3rd and 7th day.

### Isolation and culturing of FBs and ECs from the CC

The method of tissue digestion was the same as that of the single-cell sample preparation, the difference was that the digestion time for ECs was 20 min, while that of FB cells was about 40 min. The digested single-cell suspensions were washed and re-suspended in 10% FBS/DMED (Gibco) and EGM-2 (CC-3202; Lonza) medium for FB and EC culturing, respectively. Because of the strong proliferation capacity of FBs, FB cells would eventually dominate in DMEM medium, and the purity was more than 95% after passage. For ECs, cell cloning began at about day 5 of culture, then fibroblast inhibitors were added for 7–10 days. When isolated colonies began to fuse with each other, the EC colonies were marked and digested using a clonal cylinder (C7983-50EA; Sigma-Aldrich), and the EC suspension was further purified using CD31 magnetic beads (130-091-935; Miltenyi).

### Endothelial cell functional experiments for validating the prediction of CellChat analysis

Tube formation: EC cells were re-suspended in each group of medium supplemented (Supplement Table 6) at a concentration of 200/μl, and then seeded on μ-Slide (81506, ibidi) with pre-added 10 μl Matrigel (256230, Corning). After 2 hours, EC cells were fixed with PFA, and then incubated with FITC-phalloidin (G1028-50UL, Servicebio) for 30 min. Images were captured with an OLYMPUS IX83 confocal microscope.

Vascular bud length: EC were seeded in AggreWell (34411, STEMCELL) at a concentration of $3 \times 10^5$ per well, after 12 h, the cell mass were re-suspended and seeded on μ-Slide (81506, ibidi) with pre-added 10 μl Matrigel (256230, Corning) for 6 h. Next, cell supernatant was replaced with conditioned medium (Supplement Table 6) for 24 h and fixed. The ratio of cell mass diameter to the distance from the centre of the cell mass to the farthest end of the vascular bud in each group was recorded and counted.

Endothelial permeability: The EC were inoculated into the upper chamber of Transwell at a concentration of $2 \times 10^5$ per well, and then 700 μl EGM-2 (CC-3202; Lonza) medium was added into the lower well. After cell adherence, the supernatant of the upper well was replaced with conditioned medium of each group (Supplement Table 6), and cultured for 24 h. Then 70-kda TRITC-dextran (T1162, Sigma-Aldrich) with a final concentration of 2 mg/ml was added into the upper well, 5 h later, the fluorescence intensity at 590 nm of the liquid in the lower well was detected.

### RNA-seq of FB in vitro

$1 \times 10^5$ FB cells were seeded in a six-well microplate containing 2 ml 10% FBS/DMEM medium per well. After 24 h, the medium was replaced by 2% FBS/DMED containing 10 UM of ICG-001 (S2662, Selleck), 10 UM of SKL2001 (S8334, Selleck), or an equal volume of DMSO. The medium was changed every other day during culturing. After 7 days, all cells were harvested with TRIzol (15596026; Thermo Fisher) for mRNA sequencing. Library construction and sequencing were performed by Sinotech Genomics Co., Ltd (Shanghai, China). Total RNA was isolated using RNeasy mini kit (74904; Qiagen). The poly(A)-containing mRNA molecules were purified using poly(T)-oligo-attached magnetic beads. Then the mRNA underwent fragmentation, reverse transcription, purification, and enrichment. Clusters were generated by cBot with the library diluted to 10 pM and then were sequenced on the Illumina NovaSeq 6000 (Illumina, USA). Paired-end sequence files (fastq) were mapped to the reference genome (GRh38-1.2.0) using HISAT2. Differential expression analysis for mRNA was performed using the R package edgeR. Differentially expressed RNAs with $|\log_2(FC)|$ value >1 and $q$ value <0.05, considered significantly modulated, were retained for volcano plot and GO analysis.

### Quantitative real-time PCR

For quantitative real-time PCR, total RNA was isolated using the TRI-zol and chloroform liberation method. First-strand cDNA synthesis was done using the First Strand cDNA Synthesis Kit (K1651; Thermo Fisher). Quantitative PCR reactions were performed using Power SYBR® Green PCR Master Mix (4367659; Applied Biosystems) and a 7500 Fast Real-Time PCR System (4484643, Applied Biosystems). The primer pairs used are listed in Supplementary Fig. 9. PCR products were qualified by the comparative Ct (threshold cycle) method and the copy number for each transcript was expressed relative to that of the housekeeping gene *ACTB*. The software used for collecting qRT-PCR data was Bio-Rad CFX manager 3.1.

### Statistics and reproducibility

Statistical significance was calculated by GraphPad Prism 8 software with Tukey's multiple comparisons test of ANOVA (for the result in Supplementary Fig. 3d, Supplementary Fig. 4b, c, g, j, Supplementary Fig. 5a, c, e, Fig. 3e, Supplementary Fig. 10, Supplementary Fig. 11b, f, Figs. 5g, 6b, e, g and Supplementary Fig. 17d, unpaired, two-tailed) and ordinary one-way ANOVA (Supplementary Fig. 1b, d). The confidence interval was 95%. Results were considered significant at $P < 0.05$. Statistical analysis between one group and the rest of the scRNA-seq data (for the result in Supplementary Data 3–5, 8, 11 and 13) were made by the statistical analysis was made by Wilcoxon (Mann–Whitney) rank-sum test; two-tailed, with the Seurat package in R, and the confidence interval is 95%. Statistical parameters are reported in the respective figures and figure legends. Masson and IHC staining for each disease type (for the result in Figs. 4c, 5d–f, 6b, g, h; Supplementary Figs. 1b, 3c, 8c, 9a–f, 12b, 13c, d, 16d, 17c) in this study were based on at least two biological repeats. Scanning electron microscope and transmission electron microscope image (Fig. 6a, e) were based on two biological repeats.

### Reporting summary

Further information on research design is available in the Nature Research Reporting Summary linked to this article.

## Data availability

All scRNA-seq and bulk RNA-seq raw data generated for this study have been deposited in the National Omics Data Encyclopedia

database repository "OEP002406", Genome Sequence Archive "PRJCA009769" and GEO database "GSE206528". The cell–gene matrix has been uploaded in "GitHub [https://github.com/zlyingithub/single-cell-sequence-of-human-corpora-cavernosa]". All other relevant data supporting the key findings of this study are available within the article and its Supplementary Information files or from the corresponding author upon reasonable request. Source data are provided with this paper.

## Code availability

All code associated with this manuscript and the gene annotation list have been uploaded to "GitHub [https://github.com/zlyingithub/single-cell-sequence-of-human-corpora-cavernosa]" and "Zenodo [https://zenodo.org/record/6596754#.YqNdy3ZBwxY]".

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

## Acknowledgements

This work was supported by grants from Shanghai Scientific and Technological Project (21410760300, Z.L.); China Postdoctoral Science Foundation (2021M703747, L.Z.); GuangDong Basic and Applied Basic Research Foundation (2021A1515111109, L.Z.); National Natural Science Foundation of China (82171597, Z.L.; 81871215, Z.L.; and 82071636, Y.T.); the open funds of Guangdong Provincial Key Lab of Biomedical Imaging (GPKLBI202104, L.Z.). We wish to thank the Sinotech Genomics CO. LTD. Shanghai for their support of RNA-seq library preparation. We thank Dr. Jianming Zeng and his bioinformatics team for their reference codes. We thank Catherine Perfect, MA (Cantab), from Liwen Bianji (Edanz) (www.liwenbianji.cn/), for editing the English text of a draft of this manuscript.

## Author contributions

L.Z., Z.L., D.W., J.L. and T.L. designed the experiments. H.S., E.Z., P.L., C.Y., R.T., H.C. (HuiXing Chen), J.H. and H.C. (Huirong Chen) collected the samples and clinical information. Z.L., J.L. and C.S. performed immunostaining. Z.J., G.W., W.Z., and L.Z. isolated cells for scRNA-seq. Z.L., J.L. and S.H. performed cell culture experiments. L.Z. and Y.T. performed bioinformatics analyses. G.L. interpreted data. Y.C. drew the Schematic illustration. L.Z., Z.L. and T.L. wrote the manuscript. All authors revised and approved the manuscript.

## Competing interests

The authors declare no competing interests.
