## [Peer Review File · Nature Communications]

REVIEWER COMMENTS

Reviewer #1 (Remarks to the Author):

The human corpus cavernosum (CC) represents a unique vascular structure with distinct physiological properties. As observed in many tissues, increasing age and other health conditions compromise function and this is typically associated with a change in cellular composition with increased fibrosis. To better understand the cellular basis for these properties, Zhao and colleagues have generated an scRNA-seq atlas of the various cells found within the corpus cavernosum in healthy individuals (n=3), those with erectile dysfunction (n=3) and diabetes (n=2). Numerous populations were identified and characterized including mesenchymal/fibrogenic, endothelial and associated cell types (i.e., smooth muscle cells, etc.). These analyses yielded a variety of clusters with an underlying mesenchymal/fibrogenic signature which the authors collectively referred to as mesenchymal regulatory fibroblasts (MRFs) based, in part, on observed signaling activity. Furthermore, to substantiate the claim that the authors have identified a new mesenchymal cell type (line 568), much more evidence needs to be provided. However, the MRF term also generates additional confusion, as fibroblasts exhibit many of the properties identified with MRFs and it is unclear based on the analyses if these cells represent a specialized type of fibroblast. Numerous excellent human fibroblast datasets exist (Buechler et al., Nature, 2021 along with other publications) and it would be interesting to compare these cell types across tissues, to determine if the CC contains a specialized fibroblastic population. Of particular interest, is how these populations may contribute to CC dysfunction, and this is not adequately addressed in the paper. From what I can ascertain, all the analyses were carried out on aggregated data (normal, ED and diabetic) and ideally a baseline would be established with the normal samples with subsequent comparison to the "diseased samples".

Major

1) As noted above, the data across all samples is aggregated and this also appears to be used to infer what is happening in the normal state. For instance, in Figure 1e, would the identified cell-cell communication signaling interactions identified in the normal state (3 normal samples) be altered in either of the two ED states. This type of analysis applies to many of the figures in the manuscript, where healthy and diseased samples have been aggregated. If you carefully examine, Figure 1b containing the complete aggregate, can robust new clusters or altered "transcriptomic" states be identified between normal and diseased. Instead of introducing new terminology for what appear to be fibroblasts, and introducing additional terminology into the literature it would be ideal to place these cells in the framework of the current fibroblastic literature, and some of the markers being used to define these populations across tissues, such as DPT, COL15a1, PI16, PDGFRA, etc.

2) Many of the populations were identified using well expected markers and for the most part is consistent with what would be expected, with the exception of pericytes. There should be an abundant pericyte population, defined by markers such as RGS5, KCN8 and other markers. There is considerable overlap between the signatures of pericytes and smooth muscle cells, however, the latter can be typically distinguished from the former by the expression of MYOCD. There are a number of great papers from the Betsholtz's group that delves into these populations. The MRF4 cluster containing KERA, THBS4, SCX is consistent with a tenogenic signature which should be mentioned. The myofibroblasts can also likely be separated into at least two categories, based on the expression of various genes encoding extracellular matrix proteins and proteins associated with their generation (COL1A1, POSTN, LOX, LOXL1, etc.). It would be expected that the more fibrotic samples should contain an increased abundance of ECM-producing myofibroblasts, assuming that

the tissue dissociation conditions can effectively liberate these more-ECM embedded cells.

3) In Figure 3c, MRF1 exhibits increased expression of FOS, JUN, EGR1, and the authors use this observation to support their contention that the MRFs represent regulatory signaling cells. These genes are often up-regulated during tissue dissociation, and they can also exhibit cluster-specific increased expression. This is often an artifact of tissue dissociation.

4) In line 258-285, the authors suggest that canonical WNT pathway activation drives the myofibroblast phenotype, however, this appears to be mostly correlative. To support this possibility a variety of in vitro experiments are carried out with inhibitors and agonists. However, the data in Supp. 4e with canonical WNT pathway agonists and antagonists doesn't support modulation of the myofibroblastic phenotype. Standard fibrosis genes should be assessed (COL1A1 and 1A2, LOX, POSTN, etc.) and morphology is not a suitable proxy (Figure 3e and 3f) for a more thorough transcriptomic (or immunodetection) analysis of the resulting phenotypes in the treated cultures. Moreover, presumably the cultures contain a heterogeneous mixture of cells, which likely also confounds these analyses. In supplemental 5f, how many replications were performed and what number of independent biological samples were assessed. The shown genes in Supplemental 5e don't fit with the phenotypes, genes such as COL1A1 and others being "down-regulated" under both treatment conditions.

5) The data suggesting MRFs differentiate into myocytes is very weak. What is more likely, is that the cultures are contaminated with pericytes which do have the potential to become vascular smooth muscle. I would also be careful with the term myocyte and be specific on which type of myocyte you are referring (i.e., smooth, striated/skeletal, etc.). Similarly, for the proposed experiments assessing putative YAP-signaling modulators on fibrosis, it is unclear why the authors switched from primary cultures to cultured "corpora cavernosa tissue" (line 1236). I couldn't find an explanation for the methodology used in Supplemental Figure 8D. Additional data would be needed to support the claim that "inhibition of YAP alleviated EC injury in vivo" (line 1221).

6) The evidence supporting conversion of endothelial cells to a mesenchymal phenotype (EndoMT) following treatment with TGFbeta needs further validation. EndoMT is often seen in vitro (assuming one is working with non-fibroblast contaminated cultures) with TGFbeta treatment and this typically doesn't hold up in vivo.

Minor comments:

In the text with the first occurrence of Pi16 it would be useful to provide context and a reference.

How was the apparent increase in the MRF1 sub-cluster in ED patients confirmed.

It is not clear on lines 340-346 where various GO functions are listed how this impacts our understanding of MRF1 and MRF2.

For Figure 6F it is not clear how the sample numbers are generated n=10, 30, etc. if only 8 samples were used initially. Furthermore, there is no legend for 6F described. It is unclear if the n represents independent biological samples vs multiple samples from the same

specimen. In general, sample number and how it was determined needs to be added to numerous figure legends.

The discussion is mostly a rehash of the results and this can be streamlined.

In Figure 5d, there appears to be a significant amount of background with the anti-KIT antibody. It may be useful to include controls to demonstrate specificity, likewise for the anti-GJA5 staining.

Carefully check information in figures, $INF\gamma$ in Figure 1d and Supp. 5d, presumably should be $INF\gamma$. Figure 3g, legend structure should be structure. Are the vertical lines in the heatmaps in figures 2c, 4b and 5b supposed to line up with clusters? In Figure 2f, is there supposed to be p values associated with each of the violins. I would remove the 3 p-values shown.

For the identification of the "endothelial fibroblast", are there unique markers that are enriched in this population or is it truly a blend of both cell types. We have encountered this sort of population in some tissues, however, due to the absence of unique and/or definitive markers, we remain skeptical of their authenticity. A word of caution. Both populations are closely apposed in tissues, and thus, it is sometimes challenging to adequately dissociate these cell types.

Figure 6A, presumably the EM images shown are representative of what is observed in these conditions.

Eight samples, including two diabetes samples, are listed in Figure Supp. 1A, but only 7 are referred to in the text (line 565), but this appears to be an oversight and likely it should be eight. T

Grammar and spelling need attention throughout the manuscript. In most instances where pair-wise T-tests were performed it would be more appropriate to carry out ANOVA with post-hoc analyses.

Reviewer #2 (Remarks to the Author):

Erectile dysfunction(ED) is a common manifestation of aging, diabetes, hypertension, and hypogonadism — and an untoward side effect of many medications. The multifactorial physiological and psychological contributions to ED give rise to an increasingly common and clinically vexing problem for our patients, especially those with diabetes. The cellular heterogeneity of the vascular compartments and components necessary for healthy penile blood flow and successful erection are poorly understood. In this interesting manuscript, Zhou and colleagues have undertaken sc-RNA-seq assessment of penile corpus cavernosa (CC) tissue from 3 health men and 4 with ED including one man with diabetes (T1D vs. T2D and duration of disease not specified). Increased histological evidence of fibrosis was observed in men with ED, characterized by concomitant abnormalities in endothelial cell DNA damage response (γ H2AX) and evident of endothelial-mesenchymal transitioning (EndMT). Global transcriptomic analysis of CC cells by sc-RNA-seq identified ECs, fibroblasts (MRFs), macrophages (MACS), SMC, Schwann cells, and T cells — with endothelial fibroblasts and myofibroblasts undergoing phenotypic modulation from ECs and SMC, respectively, as intermediates. CellChat analyses of receptor-ligand, receptor-matrix, and cell-cell interactions identified 115 significant interactions, mostly "outgoing" in predicted vector from the MRF populations. Macrophages, ECs, and Schwann cells emerged

as targets of these signals, while T cells were not significantly mapped as engaged in this analysis. MRFs were identified to exhibit significant heterogeneity that fell into 5 sub clusters, reality defined by patterns of collagen gene expression. IHC identified spatial segregation of these MRFs into 5 distinct histoanatomic reason. MRF1 populations were increased in non-DM patients vs. ED patients and noted to be elevated in the patient with diabetes and ED. In addition to immediate early gene responses (Fos, jun, Egr1) in MRF1, Ctnnb1 and Myc er increased primarily in the MRF1 cluster of ED patients. Treatment with ICG-001, and inhibitor of Wnt signaling antagonist of CBP co-activator engagement, reduced myofibroblast fibrotic responses to regulatory fibroblasts / MRFs. the SMC lineage including myofibroblasts undergoing phenotypic modulation also exhibited heterogeneity with cell death program evidence in SMC1 but proliferative responses noted in SMC2 and SMC4 representing another myofibroblast based upon transcriptomic clustering and synthetic-contractile phenotypic characteristics. Dynamic expression of adrenoreceptors were noted between the diverse SMC sub clusters with ED. Foxo1 and Hippo pathways appeared to be downregulated in SMC cells of ED patients. Predications are that Yap1 and Taz/Wwtr1 nuclear targets would be upregulated with the latter alteration in SMC1, SMC2, and SMC4 cells. One such target, Ctgf was upregulated in ED patients, while other prototypic targets were not interrogated. However, in ECs, increased Yap1 nuclear accumulation was noted as consistent with this, but SMC localization was not described. Treatment of CC tissue ex vivo with XMU-MP-1— antagonist of the Yap1 inhibitory MST1/2-LATS1/2 relay — phenocopied the profibrotic responses observed in ED. Conversely, Yap1 pathway inhibitors, Verteporfin and TED-347 and peptide 17, reduced CC fibrotic characteristic in culture. Thus the authors conclude that these studies identify cellular targets and signaling pathways in the human CC microenvironment that serve as a roadmap / atlas for therapeutic intervention for ED therapy.

While this manuscript represents an important contribution — a roadmap / data atlas that generates hypothesis for novel approaches to ED therapeutics — it lacks robust experimental design stemming from mechanistic insight that demonstrates unambiguously that the pathways identified are pivotal in disease biology. TGF-beta, Wnt / beta-catenin and Yap/Taz signaling figure prominently in many wound healing and contractile responses. The Yap signal activation observed in the EC clusters as better resolved / defined by nuclear Yap and reduced (cytoplasmic) phosph-Yap1 is very likely to be important in the SMC clusters as well. These will almost certainly emerge as important contributors worthy of investigation and targeting, but did not required sc-RNA-seq data to be implicated in the biology. The comparisons to a signal man afflicted with ED and DM are not helpful, since the many clinical confounders (T1D vs. T2D, duration, presence of autonomic neuropathy, Tobacco history, any CKD, medications, etc) render those findings of uncertain significance in the absence of replication or validation in a preclinical model. The most intriguing aspects of this study relate to the distinct histoanatomical spatial patterns of SMC, MRF, and ECs subtypes identified that dynamically change with disease and with unique vectors of signaling from instructive MRF signals. However, experimental design was not developed that demonstrated in orthogonal assays that these interactions predicted from CellChat analysis were actually occurring and functionally important. In the absence of supportive experimental design that better demonstrates mechanistic relevance to the paracrine biology proposed for progression of ED, the manuscript in its present form does not appear rise to the level of significance routinely required for publication in the journal. A subspecialty urology journal may be another option.

Reviewer #3 (Remarks to the Author):

The authors had profiled 64993 individual penile cavernosal cells from there men with normal erections and five patients with organic ED (biopsy samples in artificial cavernous

body implantation; two DMED, three non-DMED) with single-cell sequencing analysis. As the authors have stated, these data provide insights into the cellular composition, biological features, and regulatory signalling networks of the corpora cavernosa, to give a better understanding of the human corpora cavernosa microenvironment and its dynamic changes in ED patients. It is also an interesting and very informative article for potential ED therapies.

Q1. Diabetic erectile dysfunction is a combination of microvascular, macrovascular, endocrine, and neurological disorders, and the authors seem to have described most of the etiologies. However, whether the angiogenic factors, neuroregenerative factors, and endocrine-related signaling pathways were also altered? Especially in normal and DMED patient.

Q2. Diabetic ED is known to be associated with severe structural damage, if possible, please add the cell type (cluster) ratio for the eight major cavernosal cell types from normal and ED patients.

Q3. Please provide more detailed information for subjects included in the study.

- IIEF EF or IIEF-5 scores for all subjects**
- Please provide previous ED treatment modality, such as oral PDE5I or ICI for 5 patients with ED**
- Two patients with diabetic ED: please provide data for HbA1c, anti-glycemic medication, etc**
- Three patients with non-diabetic organic ED; what is exact cause for ED? More detailed information, such as vascular risk factors or neurological causes, are needed.**

Q4. The characteristics of organic ED are somewhat not homogenous, i.e., 2 patients with diabetic ED and 3 patients with non-diabetic organic ED. I think this is a major limitation of single cell transcriptome analysis used in this study.

Reviewer #1

The human corpus cavernosum (CC) represents a unique vascular structure with distinct physiological properties. As observed in many tissues, increasing age and other health conditions compromise function and this is typically associated with a change in cellular composition with increased fibrosis. To better understand the cellular basis for these properties, Zhao and colleagues have generated an scRNA-seq atlas of the various cells found within the corpus cavernosum in healthy individuals (n=3), those with erectile dysfunction(n=3) and diabetes (n=2). Numerous populations were identified and characterized including mesenchymal/fibrogenic, endothelial and associated cell types (i.e., smooth muscle cells, etc.). These analyses yielded a variety of clusters with an underlying mesenchymal/fibrogenic signature which the authors collectively referred to as mesenchymal regulatory fibroblasts (MRFs) based, in part, on observed signaling activity. Furthermore, to substantiate the claim that the authors have identified a new mesenchymal cell type (line 568), much more evidence needs to be provided. However, the MRF term also generates additional confusion, as fibroblasts exhibit many of the properties identified with MRFs and it is unclear based on the analyses if these cells represent a specialized type of fibroblast. Numerous excellent human fibroblast datasets exist (Buechler et al., Nature, 2021 along with other publications) and it would be interesting to compare these cell types across tissues, to determine if the CC contains a specialized fibroblastic population. Of particular interest, is how these populations may contribute to CC dysfunction, and this is not adequately addressed in the paper. From what I can ascertain, all the analyses were carried out on aggregated data (normal, ED and diabetic) and ideally a baseline would be established with the normal samples with subsequent comparison to the “diseased samples”.

Response to Reviewer #1 (you can quickly view each part through the catalog):

Thank you for your suggestions. In this study, we noticed that the FB cluster (which was labeled as MRF in the last version) showed some mesenchymal characteristics such as CD105, CD73, CD90, CD29, and CD44 (Figure R1-1a, 1b). Except for the marker expression, our further study indicated that these cells have multilineage differentiation ability (Figure R1-1c). However, the differentiation efficiency was low and about only 5~10% CC fibroblasts cells were induced differentiation successfully in vitro, so, we thought even these fibroblasts exhibit many of the properties identified with mesenchymal cells, the FB standardized name was helpful in reducing confusion for readers before more evidence was given. In the revised manuscript, we combined this cluster with other fibroblasts datasets, and use published classification criteria to rename our 6 subclusters of CC fibroblasts (Buechler et al. 2021).

Figure R1-1. (a-b): the tSNE plot and mesenchymal cell markers expression of 8 clusters in the last version. (c): multidirectional differentiation experiment of CC fibroblast *in vitro*, the three panels from left to right were safranin O staining, oil red staining and alcian blue staining.

Another your concern is “the analyses were carried out on aggregated data”. In the revised manuscript, we first establish a normal baseline for the FB, SMC, and EC cluster respectively, and the heterogeneity within each major cluster (such as spatial localization and subcluster transcription characteristics) were described under health conditions based on 3 normal samples. Then we compared non-DM and DMED with normal in each subcluster, the DEGs and functional enrichment were added as Supplementary Figures and tables in our revised manuscript. What’s more, the difference in cell-cell signaling interaction between normal and patients was also added (Figure R1-2).

In addition, some other revisions made according to two other reviewers include: (1) A more detailed clinical information of these samples was added (the Supplementary Table 1). (2) IHC staining of YAP/p-YAP with SMA protein to observe the change of YAP signaling in CC SMC (corpus cavernosal trabecular smooth muscle cells) were added (the Supplementary Figure 13). (3) We validated CellChat’s predictions by expression and function experiment *in vitro* (the Supplementary Figure 3-5 and the Supplementary table 6). Thank you for your advice, which makes the logic of the article more reasonable.

Major comment 1

As noted above, the data across all samples is aggregated and this also appears to be used to infer what is happening in the normal state. For instance, in Figure 1e, would the identified cell-cell communication signaling interactions identified in the normal state (3 normal samples) be altered in either of the two ED states. This type of analysis applies to many of the figures in the manuscript, where healthy and diseased samples have been aggregated.

If you carefully examine, Figure 1b containing the complete aggregate, can robust new clusters or altered “transcriptomic” states be identified between normal and diseased. Instead of introducing

new terminology for what appear to fibroblasts, and introducing additional terminology into the literature it would be ideal to place these cells in the framework of the current fibroblastic literature, and some of the markers being used to define this populations across tissues, such as DPT, COL15a1, PII16, PDGFRA, etc.

Response: Thank you for your comment, we have deleted the cell-cell interaction analysis base on the mix of three types of samples and added a comparison between normal and non-DM/DMED to analyze what signaling may cause the pathological injury in DMED CC (Figure R1-2, it is also added as the new Supplementary Figure 2 in the revised manuscript).

Figure R1-2 (Supplementary Figure 2 in the revised manuscript). Cell-cell communication network within the

corpora cavernosa microenvironment in normal and pathological state. (a) The barplot showed the total interactions count and strength among normal, non-DM and DMED CC. (b) The networks showed the total interactions count and strength with 7 major clusters among normal, non-DM and DMED CC. (c) The heatmap showed the outgoing signaling patterns between normal and DMED CC, the barplot on the right of heatmap indicated the outgoing strength. (d) Circle plots (by CellChat analysis) depict the differential strength (right) of specific signaling in the cell-cell communication network between normal and DMED CC. Red or blue edges represent increased or decreased signaling in normal state.

Furthermore, the interaction pattern among 7 major clusters was also updated as the new Figure 1e in the revised manuscript (Figure R1-3). The FB cluster was still the most powerful outgoing signaling source.

Figure R1-3 (Figure 1e in the revised manuscript). Cell-cell communication signalling network among the 7 major clusters analysed with CellChat. the right panel showed the two-dimensional sort of 7 major clusters according to their total incoming and outgoing strength.

To functionally confirm interactions predicted from CellChat analysis (suggestion of reviewer #2), we chose some ligands which showed top different relative information flow between normal and DMED CC to perform single factor stimulation and orthogonal assays (the results have been added as the Supplementary Figure 3-5 in the revised manuscript). And we identified some factors beneficial and detrimental to the structure and function of the CC according to functional experiments.

Another concern in your major comment 1 was “If you carefully examine, Figure 1b containing the complete aggregate, can robust new clusters or altered “transcriptomic” states be identified between normal and diseased”. In order to meet this concern, we split the FB cluster (and each subcluster) according to their sample type and compared the DEGs and functional annotations between normal and DMED patients (Figure R1-4 and R1-5, R1-5). Similar operations were performed in SMC and EC cells (in the Supplementary Figure 14-15 and the Supplementary Table 12 and 14 in the revised manuscript).

Figure R1-4. the DEGs and functional annotations between normal and non-DM CCFB. (a) Violin plot showing the different expression of top DEGs between normal and non-DM CCFB. (b-h) Heatmap showed the DEGs between normal and non-DM in all CCFB or each CCFB subclusters, the right panel showed the GO or KEGG enrichment terms.

Figure R1-5 (Supplementary Figure 7 in the revised manuscript). the DEGs and functional annotations between normal and DMED CCFB. (a) Violin plot showing the different expression of top DEGs between normal and DMED CCFB. (b-h) Heatmap showed the DEGs between normal and DMED in all CCFB or each CCFB subclusters, the right panel showed the GO or KEGG enrichment terms.

Lastly, to meet the concern “introducing additional terminology into the literature it would be ideal to place these cells in the framework of the current fibroblastic literature”, we download the datasets from 9 different studies (Table R1-1) including heart(Rao et al. 2021), skeletal muscle(De Micheli et al. 2020), colon(Kinchen et al. 2018), liver(Ramachandran et al. 2019), kidney(Kuppe et al. 2021), skin(Deng et al. 2021; Theocharidis et al. 2022), and lung(Reyffman et al. 2019; Habermann et al. 2020). Fibroblasts were identified by the expression of PDGFRA, LUM, DPT and

DCN. The standardized naming of FB subcluster across tissues referenced Buechler's study (Buechler et al. 2021), however, considering CC was a specific type of vascular structure, additional dataset of heart, muscle, skin (contain small vessels) and kidney were also added (Figure R1-5, it is also added as the new Supplementary Figure 6 in the revised manuscript). Except for some clusters which have been annotated by Buechler, a novel cluster (APOC1+FB) enriched in CC, heart, and muscle was identified. The similarity between fibroblasts derived from each tissue was compared and we found that CCFB was more similar to that of heart and skeletal muscle (Figure R1-6c). Under this classification criteria, our fibroblasts are divided into 3 major clusters (PI16+, APOC1+, and COMP+), and further into 6 subclusters based on some CCFB specific markers including APOC1+/PTCHD1+, APOC1+/PPP1R14A+, PI16+/FMO2+, PI16+/BMP7+, COMP+/KERA+ and COMP+/MFAP5+ FB (Figure R1-7, it is also added as the new Figure 2a in the revised manuscript).

Study	Data Source	Disease	Platforms
this paper	this paper	Erectile function	NovaSeq6000
Rao, M., et al. (2021)	GSE145154	Dilated/Ischaemic cardiomyopathy	HiSeq X Ten
De Micheli, A.J., et al. (2020)	GSE143704	None	NextSeq 500
Kinchen, J., et al. (2018)	GSE114374	Inflammatory bowel disease	hiSeq4000
Ramachandran, P., et al. (2019)	GSE136103	Liver cirrhosis	hiSeq4000
Kuppe, C., et al. (2021)	Zenodo.4059315	Chronic kidney disease	NovaSeq6000
Deng, CC., et al. (2021)	GSE163973	Keloid	NovaSeq6000
Theocharidis, G., et al. (2022)	GSE165816	Diabetic foot ulcers	NovaSeq6000
Reyfman, PA., et al. (2019)	GSE122960	Pulmonary fibrosis	hiSeq4000
Habermann, A. C., et al. (2020)	GSE135893	Pulmonary fibrosis	NovaSeq6000

Table R1-1 the information of fibroblastic literature referred in this study.

Figure R1-6 (Supplementary Figure 6 in the revised manuscript). Cross-tissue organization of the fibroblast. (a) tSNE plots of all fibroblast from 8 tissues under normal and disease state. (b) Bar plot showing the cell count proportion of each FB subcluster in different sample types. (c) Bubble diagram showing the similarity (correlation) of FB among different sample types. A gradient of light blue to red indicates the P-value. The size of the bubble indicates the correlation coefficient. (d) Heatmap of the top 30 DEGs in each FB from different sample types. A gradient of light blue to dark red indicates low to high expression levels in the heatmap. DCM: dilated cardiomyopathy; ICM: ischemic cardiomyopathy; IPF: idiopathic pulmonary fibrosis; IBD: inflammatory bowel disease; CKD: chronic kidney disease; DFU: diabetic foot ulcer.

Figure R1-7 (Figure 2 in the revised manuscript). The 6 subcluster of CCFB from normal male and the expression pattern of their marker genes.

Major comment 2

Many of the populations were identified using well excepted markers and for the most part is consistent with what would be expected, with the exception of pericytes. There should be an abundant pericyte population, defined by markers such as RGS5, KCN J8 and other markers. There is considerable overlap between the signatures of pericytes and smooth muscle cells, however, the latter can be typically distinguished from the former by the expression of MYOCD. There are a number of great papers from the Betsholtz's group that delves into these populations. The MRF4 cluster containing KERA, THBS4, SCX is consistent with a tenogenic signature which should be mentioned. The myofibroblasts can also likely be separated into at least two categories, based on the expression of various genes encoding extracellular matrix proteins and proteins associated with their generation (COL1A1, POSTN, LOX, LOXL1, etc.). It would be expected that the more fibrotic samples should contain an increased abundance of ECM-producing myofibroblasts, assuming that the tissue dissociation conditions can effectively liberate these more-ECM embedded cells.

Response: Thank you for your advice, we use 3 cardiac markers (PLN, MYOCD, and CRYAB) and 6 pericytes markers (PDGFRB, KCN J8, ABCC9, LHFP, ANGPT2, RGS5, and KCN J8) to define the pericytes in the revised manuscript (Figure R1-8). These markers were according to Betsholtz's group studies(Armulik et al. 2011; Vanlandewijck et al. 2018; Mäe et al. 2021) and CellMarker database(Zhang et al. 2019). In the last version, we named this cluster of cells as SMC3 and find a high expression of THY1. Base on the IHC result, we found this THY1^{high} cells were located around small vessels (we mistakenly neglected this region in the last version) but not in cavernosal trabecular (their relationship is similar to heart muscle and coronary artery). In addition, the tenogenic signature of FB4 (MRF4) has been added in the revised manuscript. Although muscle cells of non-DM and DMED patients expressed more ECM genes overall, however, we failed to further divide SMC4 (myofibroblasts) into two categories according to ECM genes, these results were added in the revised manuscript.

Figure R1-8 (Figure 1b and Supplementary Figure 12a in the revised manuscript). (a-b) tSNE plots of all corpus cavernosum cells from eight donors (three normal males and five ED patients). (b) Expression patterns of the cardiac and pericytes markers for each cluster are projected on the tSNE plot of all corpora cavernosa cells. (c-d) Expression patterns of *THY1* for each cluster are projected on the tSNE plot and violin plot of myocyte cells. (e) IHC staining of *THY1* in different regions of CC tissue. a: septum pectiniforme; b: region near septum pectiniforme; c: sinusoid (cavernosal trabeculae); d: artery; e, f: nerve bundles; f: small vessels.

Major comment 3

In Figure 3c, MRF1 exhibits increased expression of *FOS*, *JUN*, *EGR1*, and the authors use this observation to support their contention that the MRFs represent regulatory signaling cells. These genes are often up-regulated during tissue dissociation, and they can also exhibit cluster-specific increased expression. This is often an artifact of tissue dissociation.

Response: Thank you for your advice, which we hadn't taken into consideration in the last version. In order to clarify whether the up-regulated transcript factors were artifacts of tissue dissociation, we perform IHC of *JUN* and *ATF3* (two up-regulated TFs in patients' CC) and found an increased nucleus location and positive cells in non-DM and DMED CC (Figure R1-9, it is also added as the new Supplementary Figure 11 in the revised manuscript). This should be related to the enhanced proliferation ability of fibroblasts under pathological conditions. In addition, all CC tissues were dissociated under the same condition including the same enzyme concentration and dissociation time, the effects of dissociation should be consistent across pathological types theoretically. However, both scRNA-seq data (Figure R1-4/5) and IHC results showed an up-regulation of *JUN* and *ATF3* in ED (especially DMED) CC. So we thought this result was not caused by tissue dissociation.

Figure R1-9 (Supplementary Figure 10 in the revised manuscript). The IHC staining of JUN and ATF3 in 3 types of CC tissues. The scale bar represents 200 μ m. The statistical analysis was made by Tukey multiple comparisons test; the confidence interval is 95%.

Major comment 4

In line 258-285, the authors suggest that canonical WNT pathway activation drives the myofibroblast phenotype, however, this appears to be mostly correlative. To support this possibility a variety of in vitro experiments are carried out with inhibitors and agonists. However, the data in Supp. 4e with canonical WNT pathway agonists and antagonists doesn't support modulation of the myofibroblastic phenotype. Standard fibrosis genes should be assessed (COL1A1 and 1A2, LOX, POSTN, etc.) and morphology is not a suitable proxy (Figures 3e and 3f) for a more thorough transcriptomic (or immunodetection) analysis of the resulting phenotypes in the treated cultures. Moreover, presumably the cultures contain a heterogeneous mixture of cells, which likely also confounds these analyses. In supplemental 5f, how many replications were performed and what number of independent biological samples were assessed. The shown genes in Supplemental 5e don't fit with the phenotypes, genes such as COL1A1 and others being "down-regulated" under both treatment conditions.

Response: Thank you for your advice, we have added qPCR results of three fibrosis genes according to RNA-seq results and found POSTN and COL4A5 were up-regulated after SKL2001 treatment,

but LOX was down-regulated in both treatment groups just like COL1A1, COL3A2, DCN and so on (Figure R1-10). So, we agree with your concern that this transformation is not typical of myofibroblasts. In the revised manuscript, we deleted this conclusion and treated the myofibroblast concept more carefully. However, SKL2001 treatment truly increased the expression of some ECM genes and muscle-related genes, and it also promoted CCFB proliferation in vitro. These results are consistent with our single-cell sequencing analysis. Therefore, the description of these phenomena is retained in the new manuscript.

Figure R1-10 (Supplementary Figure 11e in the revised manuscript). qPCR results showing the expression fold change of collagen- and muscle-associated genes in MRFs with SKL2001 or ICG-001 treatment. The gene expression levels of normal MRFs without SKL2001 or ICG-001 treatment (DMSO treated) were used as the baseline values. Data shown as mean from four technical repeats.

Major comment 5

The data suggesting MRFs differentiate into myocytes is very weak. What is more likely, is that the cultures are contaminated with pericytes which do have the potential to become vascular smooth muscle. I would also be careful with the term myocyte and be specific on which type of myocyte you are referring (i.e., smooth, striated/skeletal, etc.).

Similarly, for the proposed experiments assessing putative YAP-signaling modulators on fibrosis, it is unclear why the authors switched from primary cultures to cultured “corpora cavernosa tissue” (line1236). I couldn’t find an explanation for the methodology used in Supplemental Figure 8D. Additional data would be needed to support the claim that “inhibition of YAP alleviated EC injury in vivo” (line 1221).

Response: Thank you for your advice, cell contamination is indeed a possible cause of this experimental phenomenon, and we have considered this factor in our previous experiments. In order to verify the purity of fibroblasts, ICC staining was performed and it was found that SMA positive cells were difficult to survive and proliferate in 2~10% FBS/DMEM (although it has been reported that rat smooth muscle cells can be cultured in vitro). In addition, considering pericytes are located

around the small blood vessels, we specially selected the region of cavernous artery where small blood vessels are enriched to perform dissociation and culture. The results show that although this operation enriched the SMA positive cells in some degree, their proportion is still very low (less than 1%) (Figure R1-11). So, we think it can rule out pericytes or muscle cell contamination in this culture system.

Figure R1-11. Identification of smooth muscle and pericyte contamination. (a-b) the region inside the rectangular contains more small blood vessels. (c) The CC tissue from both regions was dissociated and cultured, and the proportion of muscle cells was determined by SMA staining. The scale bar represents 20 μm .

Another concern is “it is unclear why the authors switched from primary cultures to cultured “corpora cavernosa tissue”. We found Hippo signaling of EC in both non-DM and DMED CC was inhibited. The imbalance of YAP signaling has been reported to be associated with EC injury and fibrosis *in vivo* (Li et al. 2019; Savorani et al. 2021). So, we first explored the effect of YAP agonists and inhibitors on CCEC (Corpus cavernosal sinusoid endothelial cells) *in vitro*, and found the YAP agonists (XMU-MP-1) induced apoptosis and exacerbated the loss of endothelial markers combined with TGF β stimulation. In the last version, we found the XMU-MP-1 treatment caused CCEC from adhesion and congregated growth into mutual estrangement, which is a manifestation of endothelial barrier damage. In recent experiments, we also found that XMU-MP-1 stimulation led to the increase of endothelial cell permeability (Figure R1-12, it is also added as the new Supplementary Figure 5a in the revised manuscript), it is bad for blood retention in CC sinusoid during erection, and is one reason why happen ED. All kinds of injury in ED CC tissue eventually result in the increase of fibrosis, although the source of the fibrosis remains controversial, maybe EndoMT,

myofibroblast differentiation or increase of fibroblast proliferation, regardless of these processes. The level of fibrosis is still a simple and feasible standard for evaluating whether a signal is good or bad for CC tissue. So, we used CC tissue culture which theoretically contains all the cellular components and signaling factors in the microenvironment, to the maximum extent possible to simulate what happens *in vivo*.

We agree with your concern that the evidence for “MRFs differentiation” and “EndoMT” is still weak, and more experiments on fibrosis evaluation parameters are needed. Unfortunately, due to the unprecedented COVID-19 outbreak in Shanghai, our lab was recently shut down completely, and further evaluation of these experiments is currently difficult to complete. So, we decided to remove the two conclusions (“MRFs differentiation” and “EndoMT”), and only describe experimental phenomena and results, including XMU treatment caused endothelial cell apoptosis, the loss of the biomarkers, endothelial barrier damage, and increased fibrosis in tissue culture. These phenomena can also explain YAP signal from the side is harmful for the structure and function of the CC tissue, and verify the predicted results of our single-cell transcriptome analysis. What’s more, we have added a more detailed description of the tissue culture method in the revised manuscript.

Figure R1-12 (Supplementary Figure 5a in the revised manuscript). (a) Morphological changes of CCEC after XMU treatment for 48h. (b) Changes in CCEC permeability after different stimulation treatments. The statistical analysis

was made by ANOVA with Tukey's multiple comparisons test; the confidence interval is 95%.

Major comment 6

The evidence supporting conversion of endothelial cells to a mesenchymal phenotype (EndoMT) following treatment with TGFbeta needs further validation. EndoMT is often seen *in vitro* (assuming one is working with non-fibroblast contaminated cultures) with TGFbeta treatment and this typically doesn't hold up *in vivo*.

Response: Thank you for your advice, we agree with your concern that EndoMT following TGFb treatment needs further validation, especially *in vivo*. In addition, we found that TGFb stimulation reduces the tube formation and barrier activity of CCEC while limiting its migration, which is not consistent with EndoMT (ECs which undergo EndoMT often showed increased migration capacity). So, we removed this conclusion about TGFb caused EndoMT and only described the effects of TGFb on various physiological functions of CCEC, and discuss the relationship between these effects on the occurrence of ED and the damage of CC tissue in the revised manuscript.

Minor comments:

Minor comments 1: In the text with the first occurrence of Pi16 it would be useful to provide context and a reference.

Response: Thank you for your advice, we have added relevant explanations and references in this paragraph of the revised manuscript.

Minor comments 2: How was the apparent increase in the MRF1 sub-cluster in ED patients confirmed.

Response: We cannot judge the absolute increase number of FB1 (APOC1+/PTCHD1+) (MRF1) based on single-cell data, but their relative proportion in all FB is increased in ED patients (Figure R1-13). Another evidence was this cluster expressed a high level of JUN and ATF3, and we found an increased JUN+ and ATF3+ cells in ED patients based on IHC staining (Figure R1-9).

Figure R1-13. The proportion of each FB cluster in three types of CC samples.

Minor comments 3: It is not clear on lines 340-346 where various GO functions are listed how this

impacts our understanding of MRF1 and MRF2.

Response: Sorry for that mistake, we miswrote SMC1 and SMC2 here as MRC1 and MRC2, and we have corrected this mistake in the new edition. What's more, in order to make it easier for readers to understand the characteristics of the four groups of muscle cells, we added a more intuitive names (VSMC: vascular smooth muscle cells; CCSMC: corpus cavernosal trabecular smooth muscle cells; PC: pericyte; MFB: myofibroblast) for SMC1-4 respectively. Since we did not know what SMC1 and SMC2 were in the early analysis, so we conducted GO analysis between them. Now their positioning in CC has been clear, so this paragraph has been deleted.

Minor comments 4: For Figure 6F it is not clear how the sample numbers are generated n=10, 30, etc. if only 8 samples were used initially. Furthermore, there is no legend for 6F described. It is unclear if the n represents independent biological samples vs multiple samples from the same specimen. In general, sample number and how it was determined needs to be added to numerous figure legends.

Response: In some experiments, for example, we select multiple fields of IHC staining for statistics based on three or more biological replicates, the "n" indicates the number of fields or cells we selected to do statistics. We have added a detailed description of biological replicates and technological replicates for each figure in our revised manuscript.

Minor comments 5: The discussion is mostly a rehash of the results and this can be streamlined.

Response: Thank you for your advice, the discussion has been re-edited and some descriptions of the results have been removed in the revised manuscript.

Minor comments 6: In Figure 5d, there appears to be a significant amount of background with the anti-KIT antibody. It may be useful to include controls to demonstrate specificity, likewise for the anti-GJA5 staining.

Response: Thank you for your advice, the negative controls (use normal IgG to replace primary antibody) for these staining were added in the revised manuscript.

Minor comments 7: Carefully check information in figures, $INF\gamma$ in Figure 1d and Supp. 5d, presumably should be $IFN\gamma$. Figure 3g, legend struture should be structure. Are the vertical lines in the heatmaps in figures 2c, 4b and 5b supposed to line up with clusters? In Figure 2f, is there supposed to be p values associated with each of the violins. I would remove the 3 p-values shown.

Response: Sorry for these mistakes, we have corrected the related spelling errors. Because the DEGs number of each cluster was not the same, it is difficult for us to map the vertical line to each group. For easy reading, the left side of the vertical line is marked with the name of each cluster. The p values of Figure 2f were removed in the revised manuscript. Thank you again for your thoughtful advice.

Minor comments 8: For the identification of the "endothelial fibroblast", are there unique markers that are enriched in this population or is it truly a blend of both cell types. We have encountered this sort of population in some tissues, however, due to the absence of unique and/or definitive markers, we remain skeptical of their authenticity. A word of caution. Both populations are closely apposed in tissues, and thus, it is sometimes challenging to adequately dissociate these cell types.

Response: Thank you for your advice. In order to verify your concerns, we use the endothelial cell marker CD31 and fibroblasts markers LUM to perform IHC staining in normal and DMED CC tissues. The results showed that a group of cells, which are in the position of the endothelial cells, expressing both two markers. These cells were easier to be observed in DMED than in normal CC (Figure R1-14a). In 3D magnified field, we can find the cytoplasmic protein LUM is located medial to the membrane protein CD31 and lateral to the nucleus in the same cell (Figure R1-14b X axis magnification). In addition, the Figure 5f also showed the expression of EC marker VWF and EC4/FB marker IGF1 within one cell in DMED CC. This result confirmed the presence of endothelial fibroblasts in CC, especially in DMED CC.

Figure R1-14. The CD31 and LUM IHC staining of normal and DMED CC tissues. The X and Y axis magnification

field was the 2D view of X - or Y - axis recombined with Z - axis, respectively.

Minor comments 9: Figure 6A, presumably the EM images shown are representative of what is observed in these conditions.

Response: Your concern represented a possibility, even though we have proved endothelial fibroblasts were present in CC tissue, the interference caused by double-cells cannot be completely ruled out. It was due to the limitations of current single-cell technologies, we don't have a good way yet to know for sure whether double-cells are involved.

Minor comments 10: Eight samples, including two diabetes samples, are listed in Figure Supp. 1A, but only 7 are referred to in the text (line 565), but this appears to be an oversight and likely it should be eight.

Response: Sorry for these mistakes, we have corrected the related spelling errors.

Minor comments 11: Grammar and spelling need attention throughout the manuscript. In most instances where pair-wise T-tests were performed it would be more appropriate to carry out ANOVA with post-hoc analyses.

Response: All text spelling has been checked again by two researchers and the spelling errors have been corrected. The corresponding statistical methods have also been revised in accordance with your suggestions, thank you again.

Reviewer #2

Erectile dysfunction(ED) is a common manifestation of aging, diabetes, hypertension, and hypogonadism — and an untoward side effect of many medications. The multifactorial physiological and psychological contributions to ED give rise to an increasingly common and clinically vexing problem for our patients, especially those with diabetes. The cellular heterogeneity of the vascular compartments and components necessary for healthy penile blood flow and successful erection are poorly understood. In this interesting manuscript, Zhou and colleagues have undertaken sc-RNA-seq assessment of penile corpus cavernosa (CC) tissue from 3 health men and 4 with ED including one man with diabetes (T1D vs. T2D and duration of disease not specified).

Increased histological evidence of fibrosis was observed in men with ED, characterized by concomitant abnormalities in endothelial cell DNA damage response (gammaH2AX) and evident of endothelial-mesenchymal transitioning (EndMT). Global transcriptomic analysis of CC cells by sc-RNA-seq identified ECs, fibroblasts (MRFs), macrophages (MACS), SMC, Schwann cells, and T cells — with endothelial fibroblasts and myofibroblasts undergoing phenotypic modulation from ECs and SMC, respectively, as intermediates. CellChat analyses of receptor-ligand, receptor-matrix, and cell-cell interactions identified 115 significant interactions, mostly “outgoing” in predicted vector from the MRF populations. Macrophages, ECs, and Schwann cells emerged as targets of these signals, while T cells were not significantly mapped as engaged in this analysis. MRFs were identified to exhibit significant heterogeneity that fell into 5 sub clusters, reality defined by patterns of collagen gene expression. IHC identified spatial segregation of these MRFs into 5 distinct

histoanatomic reason. MRF1 populations were increased in non-DM patients vs. ED patients and noted to be elevated in the patient with diabetes and ED.

In addition to immediate early gene responses (Fos, jun, Egr1) in MRF1, *Ctnnb1* and *Myc* were increased primarily in the MRF1 cluster of ED patients. Treatment with ICG-001, an inhibitor of Wnt signaling antagonist of CBP co-activator engagement, reduced myofibroblast fibrotic responses to regulatory fibroblasts / MRFs. The SMC lineage including myofibroblasts undergoing phenotypic modulation also exhibited heterogeneity with cell death program evidence in SMC1 but proliferative responses noted in SMC2 and SMC4 representing another myofibroblast based upon transcriptomic clustering and synthetic-contractile phenotypic characteristics. Dynamic expression of adrenoreceptors were noted between the diverse SMC sub clusters with ED. *Foxo1* and Hippo pathways appeared to be downregulated in SMC cells of ED patients. Predictions are that *Yap1* and *Taz/Wwtr1* nuclear targets would be upregulated with the latter alteration in SMC1, SMC2, and SMC4 cells. One such target, *Ctgf* was upregulated in ED patients, while other prototypic targets were not interrogated. However, in ECs, increased *Yap1* nuclear accumulation was noted as consistent with this, but SMC localization was not described. Treatment of CC tissue *ex vivo* with XMU-MP-1— antagonist of the *Yap1* inhibitory MST1/2-LATS1/2 relay — phenocopied the profibrotic responses observed in ED. Conversely, *Yap1* pathway inhibitors, Verteporfin and TED-347 and peptide 17, reduced CC fibrotic characteristic in culture. Thus the authors conclude that these studies identify cellular targets and signaling pathways in the human CC microenvironment that serve as a roadmap / atlas for therapeutic intervention for ED therapy.

Response to Reviewer #2 (you can quickly view each part through the catalog):

Thank you for your suggestions. In this study, total 8 corpus cavernosa (CC) samples including 3 males with normal erectile and 3 non-diabetic ED, and 2 T1D caused ED patients. The onset of type 2 diabetes is usually late. When the complication of ED occurs, patients are already older and lack the desire to seek surgical treatment for ED, so we have not collected appropriate samples of this kind. In addition, we have added as detailed clinical information as possible about these patients and ED-related examinations (Supplementary Table 1). These improvements will enable readers to have a more intuitive understanding of the samples enrolled in this study.

In the revised manuscript, we focused on the validation of CellChat prediction and identified some favorable and unfavorable signaling factors for CC structural repair and erectile function (Figure R2-3 to R2-8). For your concern that "T cells were not significantly mapped as engaged in this analysis", we also revised our analysis and found that T cells are involved in CCL, CXCL and TGF β signal networks (Figure R2-3c and R2-3d).

In addition, we think it is necessary to explain some other revisions made according to two other reviewers to you here. (1) Transcriptional characteristics (including the DEGs among subclusters within one major cell cluster), spatial localization, and physiological functions of each major cluster were first built based on a normal baseline, and then the comparison between normal male and non-DM/DMED patients in each subcluster were made. (2) Pericyte, which was named as "SMC3" in the last version, was identified according to the suggestion of reviewer #1. They were found around small blood vessels in the CC tissue. (3) The nomenclature of "MRF" was abandoned, and this cluster was analyzed after integration with fibroblasts from other multiple tissues, and a more standardized nomenclature was made ("MRF" was replaced by "Fibroblast, FB"). Under this classification criteria, fibroblasts are divided into 3 major clusters (PI16+, APOC1+, and COMP+) (Buechler et al. 2021), and further into 6 subcluster based on some CCFB specific markers including

APOC1+/PTCHD1+, APOC1+/PPP1R14A+, PI16+/FMO2+, PI16+/BMP7+, COMP+/KERA+ and COMP+/MFAP5+ FB. (4) Although some evidence from cell experiments *in vitro* and *ex vivo* has been showed in the last version manuscript, we were more cautious with the results of “fibroblast differentiation into myofibroblasts” and “TGFb induced EndoMT”, and describe only the experimental phenomena.

Major comment 1

While this manuscript represents an important contribution — a roadmap / data atlas that generates hypothesis for novel approaches to ED therapeutics — it lacks robust experimental design stemming from mechanistic insight that demonstrates unambiguously that the pathways identified are pivotal in disease biology. TGF-beta, Wnt / beta-catenin and Yap/Taz signaling figure prominently in many wound healing and contractile responses. The Yap signal activation observed in the EC clusters as better resolved / defined by nuclear Yap and reduced (cytoplasmic) phosph-Yap1 is very likely to be important in the SMC clusters as well. These will almost certainly emerge as important contributors worthy of investigation and targeting, but did not required sc-RNA-seq data to be implicated in the biology.

Response: Thank you for your advice. In the revised manuscript, we added YAP/p-YAP with SMA protein IHC to observe the change of YAP signaling in CC SMC (corpus cavernosal trabecular smooth muscle cells). The IHC results clearly showed YAP signaling in non-DM and DMED CC were active, including an increased nucleus location of YAP and a decreased expression of p-YAP (Figure R2-1). This is consistent with your prediction and our previous analysis (Figure R2-1a).

According to the suggestion of reviewer #1, in order to make it easier for readers to understand the characteristics of the four groups of muscle cells, we used the new names (VSMC: vascular smooth muscle cells; CCSMC: corpus cavernosal trabecular smooth muscle cells; PC: pericyte; MFB: myofibroblast) instead of the old ones (SMC1-4), respectively.

Figure R2-1 (the Supplementary Figure 13 in the revised manuscript). (a) Bubble diagram of the top activated and inhibited IPA pathways based on the DEGs between non-DM or DMED SMCs compared with normal SMCs, we can find the Hippo signaling was inhibited in both non-DM and DMED CC. (b) Violin plot combined with box plot showing the transcription levels of YAP1 in each SMC subcluster between normal male and ED patients. (c) IHC co-staining of YAP/p-YAP (green) and SMA (red) in normal and ED corpora cavernosa paraffin sections (bottom panel). Since the nuclei of CCSMC are elongated oval, in order to more clearly observe the expression of YAP in the nuclei, we selected the short-axis (Region 1) and long-axis (Region 2) cross-sections of the nuclei for observation. The scale bar represents 200 μ m.

Major comment 2

The comparisons to a signal man afflicted with ED and DM are not helpful, since the many clinical

confounders (T1D vs. T2D, duration, presence of autonomic neuropathy, Tobacco history, any CKD, medications, etc) render those findings of uncertain significance in the absence of replication or validation in a preclinical model.

Response: Thank you for your suggestion. In the last version, the description of clinical information was too simple, which may have caused some confusion. In fact, the 8 patients enrolled in this study were screened carefully, and we tried to exclude those who were older than 55 years old. Although most penile cancer patients are older than this age, increasing age also leads to more chronic disease and other confounding factors. So, we chose 55 years old as a balance that allowed us to get enough samples, but also control for age-related interference. Other patients who had other chronic diseases or bad lifestyle habits that seriously affected their health were also excluded. For the 2 DMED patients (Patients with DMED often cannot choose surgical treatment because they are prone to post-operative infections and poor wound healing, resulting in difficulty in obtaining samples), both of them had been diagnosed with type 1 diabetes for more than 10 years. For non-DM patients, we also tried to select patients with relatively consistent symptoms according to ICI tests, they all had both inadequate blood supply and increased venous return (Figure R2-2a). **A more detailed clinical information of the 8 patients was added as Supplement Table 1 in the revised manuscript.**

a

Sample ID	ED duration (Year)	IEEF-5 score	NPTR Effective erection event/Erectile event	Tip/Base maximum average hardness	ICI PSV (left/right) cm/s	ICI EDV (left/right) cm/s	ICI RI (left/right)	ICI erection hardness
non-DM_1	5	7	0/2	Tip/Base maximum average hardness 47% and 64%, swelling 19% and 22%, lasting 2 min.	30/28	10/8	0.66/0.71	Grade 2
non-DM_2	4	9	0/8	Tip/Base maximum average hardness 59% and 41%, swelling 31% and 31%, lasting 13 min.	24/25	7/6	0.70/0.76	Grade 3
non-DM_3	3	8	0/10	Tip/Base maximum average hardness 62% and 60%, swelling 29% and 30%, lasting 7 min.	18/27	4/0	0.77/1	Grade 3
Diabetes_1	3	13	0/1	Tip/Base maximum average hardness 45% and 40%, swelling 19% and 22%, lasting 4 min.	29/47	9/15	0.68/0.68	Grade 2-3
Diabetes_2	5	11	0/1	Tip/Base maximum average hardness 52% and 34%, swelling 37% and 21%, lasting 5 min.	27/28	5/6	0.81/0.80	Grade 2

b

Figure R2-2. (a) The clinical information of the 8 patients in this study. (b) The NPTR results of the 8 patients. Normal reference value: penis dilation hardness $\geq 60\%$ and duration $\geq 10\text{min}$. Free testosterone $\geq 0.0865\text{ng/mL}$. PSV $\geq 30\text{cm/s}$, EDV $\leq 3\text{cm/s}$, RI ≥ 0.8 .

Major comment 3

The most intriguing aspects of this study relate to the distinct histoanatomical spatial patterns of SMC, MRF, and ECs subtypes identified that dynamically change with disease and with unique vectors of signaling from instructive MRF signals. However, experimental design was not developed that demonstrated in orthogonal assays that these interactions predicted from CellChat analysis were actually occurring and functionally important.

Response: CC microenvironment contains many cell types and complex signaling networks, we have identified the spatial patterns of most major clusters and its subcluster based on sc-RNA seq results combined with IHC staining. However, the validation of the signaling network predicted from CellChat is a huge project, in the revised manuscript, we divided the validation experiment into four steps. First, we screened signal molecules that differed greatly between normal and patients predicted by CellChat, and these ligand molecules were further screened by whether expressing in CCFB cluster. Secondly, RNA-seq, ICC, and ELISA experiments of CCFB were used to verify whether these ligands were indeed produced by FB cells at the RNA and protein levels. Thirdly, we chose CCEC (Corpus cavernosum sinusoid endothelial cells), one of the most important cells for the CC tissue structure and erectile function, as the target cells, and checked the effect of these candidate signal ligands on CCEC through single factor stimulation. Finally, we further clarify the function of the network formed by these signal ligands through orthogonal experiments. Advanced Glycation End Products (AGEs) were used to simulate a diabetic state. Worth mentioning is that there is no gold standard for the human CCEC function appraisal at present, so we performed verification experiment in two aspects, the one is tissue damage repair associated functions such as cell migration, tube formation, vascular budding, etc., another one is erection associated function such as endothelial permeability (increased endothelial permeability prevents blood retention in CC sinusoid to maintain erections), endothelial nitric oxide and endothelin (two most important signaling factor created by EC, and regulate relaxation and contraction of smooth muscle, respectively). Below is our detailed description of the results for each step.

In the revised manuscript, we first compared the number and strength of different signals between normal and patients in each cell cluster, then we select some signals that were sent by FB and received by EC. KIT ligand (SCF) and IGF1 showed a decreased pattern in ED patients, and most other signals showed an increased pattern (Figure R2-3, it is also the Supplementary Figure 2 in the revised manuscript).

Figure R2-3 (the Supplementary Figure 2 in the revised manuscript). Cell-cell communication network within the corpora cavernosa microenvironment in normal and pathological state. (a) The barplot showed the total interactions count and strength among normal, non-DM and DMED CC. (b) The networks showed the total interactions count and strength with 7 major clusters among normal, non-DM and DMED CC. (c) The heatmap showed the outgoing signaling patterns between normal and DMED CC, the barplot on the right of heatmap indicated the outgoing strength. (d) Circle plots (by CellChat analysis) depict the differential strength (right) of specific signaling in the cell-cell communication network between normal and DMED CC. Red or blue edges represent increased or decreased signaling in normal state.

Then, we used RNA-seq, ICC and ELISA experiments of CCFB to verify that these ligands were indeed produced by FB cells at the RNA and protein levels. In ELISA experiments, we also added CCEC as a control, but CCEC cannot survive for 24h in 0% FBS condition, so we used 2% FBS DMED for culture. The results show that most of these ligands are also present in serum. CCEC culture consumed these ligand molecules, and CCFB culture increased most of these factors' concentrations (Figure R2-4, it is also the Supplementary Figure 3 in the revised manuscript). These experimental results confirm CellChat's prediction of the expression patterns of these signaling ligands.

Figure R2-4 (the Supplementary Figure 3 in the revised manuscript). The expression pattern of signaling ligands in CCFB. (a-b) RNA-seq of CCFB and CCEC that were cultured in vitro showed the expression levels of relevant signal ligands and receptors, respectively. (c) ICC staining of signal ligands predicted by CellChat in CCFB. The scale bar represents 20 μm . (d) ELISA showed the concentration change of signal ligands in CCEC and CCFB supernatant. The statistical analysis was made by ANOVA with Tukey's multiple comparisons test; the confidence interval is 95%.

Next, we evaluated the effect of single signal ligand stimulation on CCEC functions, which were related to the repair of CC tissue after pathological injury. In general, IGF1 had a positive effect on endothelial cell migration (Figure R2-5a and R2-5c), tube formation (Figure R2-6a and R2-6c), and vascular budding formation (Figure R2-7a and R2-7c), while TGF β acted as a negative regulator. This result was also verified by orthogonal experiments with various variables (Figure R2-5-R2-7 b and d, and Supplementary Table 6 Orthogonal experimental statistics results). Although the orthogonal experiment results showed the optimal ratio of signal ligand in different functional experiments was not same, in general, AGEs at low concentration with medium concentration of SCF, medium concentration of FGF7, medium concentration of VEGFA, high concentration of IGF1, low concentration of HBEGF, high concentration of CXCL12 and low concentration of TGF β ratio are more likely to facilitate the CCEC function in the repair of CC tissue injury. In addition, in order to further explore the effect of the comprehensive signaling networks released by CCFB, we also directly stimulated CCEC with FB (treated with or without AGEs) supernatant. The results showed that both supernatants promoted cell migration, and the supernatant of FB treated without AGEs also promote tube formation of CCEC. Due to the limitation of article volume, we have streamlined and integrated the data in the revised manuscript (Figure R2-5 to R2-7 were streamlined and integrated into one as the Supplementary Figure 4 in the revised manuscript).

Table R2-1. Orthogonal experimental L18($2^1 \times 3^7$) design scheme.

ng/ml	AGEs	SCF	FGF7	VEGFA	IGF1	HBEGF	CXCL12	TGF β
NO.1	10	10	100	0	10	100	10	0
NO.2	0	0	100	100	100	100	100	100
NO.3	10	100	10	0	100	0	10	100
NO.4	10	10	0	10	100	0	100	10
NO.5	0	0	10	10	10	10	10	10
NO.6	0	10	10	10	100	100	0	0
NO.7	10	100	100	10	0	10	100	0
NO.8	10	10	10	100	0	10	0	100
NO.9	0	100	10	100	10	0	100	0
NO.10	0	10	100	100	0	0	10	10
NO.11	10	0	10	0	0	100	100	10
NO.12	0	10	0	0	10	10	100	100
NO.13	0	100	0	10	0	100	10	100
NO.14	0	100	100	0	100	10	0	10
NO.15	0	0	0	0	0	0	0	0
NO.16	10	0	0	100	100	10	10	0
NO.17	10	0	100	10	10	0	0	100
NO.18	10	100	0	100	10	100	0	10

AGEs: advanced glycation end products.

Figure R2-5. CCEC migration change after single factor stimulation or the comprehensive effect of multiple factors. (a-b) Morphological characteristics of cell migration in each group under light microscope. (c-d) Statistical results of each group. The statistical analysis was made by ANOVA with Tukey's multiple comparisons test; the confidence interval is 95%.

Figure R2-6. CCEC tube formation change after single factor stimulation or the comprehensive effect of multiple factors. (a-b) Polypeptide staining showed the CCEC tube formation in each group under light microscope. (c-d) Statistical results of each group. The statistical analysis was made by ANOVA with Tukey's multiple comparisons test; the confidence interval is 95%.

Figure R2-7. The radial length of CCEC mass bud change after single factor stimulation or the comprehensive effect of multiple factors. (a-b) Morphological characteristics of cell mass bud in each group under light microscope. (c-d) Statistical results of each group. The statistical analysis was made by ANOVA with Tukey's multiple comparisons test; the confidence interval is 95%.

Another important function of CCEC is to regulate the contraction and relaxation of CCSMC. The endothelial NO-cGMP pathway is the most important signaling to promote the relaxation of the CCSMC and induce penile erection, while endothelin (ET) can effectively maintain the contraction state of CCSMC. The ELISA results indicated that most of the signal ligands even TGF β and low level of AGEs showed a positive effect on endothelial NO releasing (Figure R2-8a). However, possibly due to the synergistic or antagonistic effect within these factors, only the increase of HBEGF and CXCL12 concentration could promote the release of NO in the orthogonal experiment, and ANOVA analysis showed no statistical difference (Supplementary Table 5 Orthogonal experimental statistics results). Both high and low concentrations of AGEs can effectively stimulate the release of ET (Figure R2-8c). Except for VEGFA and CXCL12, most of the factors show inhibitory effects on ET releasing. It is worth noting that HBEGF has shown a strong inhibitory effect on ET release in both single factor stimulation and orthogonal experiment, and we thought it would be a potential drug target for promoting penile erection.

On the other hand, damage to the barrier structure composed of CCEC may prevent blood from

remaining in the cavernous sinus to maintain pressure, so we also used Transwell to detect the CCEC permeability change. The results of single-factor stimulation experiment supported the previous results and predictions of this study: high concentrations of AGEs, TGF β , and Hippo pathway inhibitor XMU-MP-1 led to increased endothelial permeability (Figure R2-8e). TGF β also showed the same effect in the orthogonal experiment, although there was no statistical difference (Supplementary Table 5 Orthogonal experimental statistics results).

Together, this additional experimental evidence suggested that signaling factors released from CCFB do exist in CC microenvironment and are functionally important. We place these results before the detailed study of CCFB (Figure 2) to suggest the importance of CCFB in the revised manuscript. Unfortunately, further validation and downstream pathways were disrupted by the recent severe COVID-19 outbreak in Shanghai. So, we added a discussion about the influence of microenvironmental signal homeostasis on the function and structure of CC in the revised manuscript based on the above experimental results and previous literature reports.

Figure R2-8 (the Supplementary Figure 5 in the revised manuscript). The change of CCEC function relate to erectile regulation after single factor stimulation or the comprehensive effect of multiple factors. (a-b) ELISA results showed

that the concentration of NO in the supernatant of EC cells cultured under various of stimulation conditions for 24 hours. (c-d) ELISA results showed that the concentration of ET in the supernatant of EC cells was cultured under various stimulation conditions for 24 hours. (e-f) Fluorescence intensity in Transwell lower pore indirectly indicated the changes of endothelial permeability under various stimulation conditions for 24 hours. The statistical analysis was made by ANOVA with Tukey's multiple comparisons test; the confidence interval is 95%.

Major comment 4

In the absence of supportive experimental design that better demonstrates mechanistic relevance to the paracrine biology proposed for progression of ED, the manuscript in its present form does not appear rise to the level of significance routinely required for publication in the journal. A subspecialty urology journal may be another option.

Response: The above updates based on your suggestions have enriched the functional depth of this study and strengthened the clinical guiding significance of our scRNA-seq data. Although our study is based on ED, a disease belonging to andrology matter, it also involves the comparison of fibroblasts (The Supplementary Figure 6) and endothelial cells (The Supplementary Figure 16b) across various tissues (Kinchen et al. 2018; Reyfman et al. 2019; Habermann et al. 2020; Kalucka et al. 2020; Buechler et al. 2021; Rao et al. 2021). What's more, ED is often considered a harbinger of cardiovascular events (Terentes-Printzios et al. 2021). They share some common organizational structures and pathogenesis. These results also have reference value for the studies of cardiovascular diseases. In addition, vascular damage caused by diabetes may occur in a variety of tissues and organs, and this paper is also helpful for the study of diabetes complications. So, we still hope to publish our research in the journal and sincerely hope to get your support, thank you.

Reviewer #3

The authors had profiled 64993 individual penile cavernosal cells from there men with normal erections and five patients with organic ED (biopsy samples in artificial cavernous body implantation; two DMED, three non-DMED) with single-cell sequencing analysis. As the authors have stated, these data provide insights into the cellular composition, biological features, and regulatory signalling networks of the corpora cavernosa, to give a better understanding of the human corpora cavernosa microenvironment and its dynamic changes in ED patients. It is also an interesting and very informative article for potential ED therapies.

Response to Reviewer #3 (you can quickly view each part through the catalog):

Thank you for taking the time to review this study. Before answering your questions, we think it is necessary to explain some other revisions made according to two other reviewers to you here. (1) Transcriptional characteristics (including the DEGs among subclusters within one major cell cluster), spatial localization, and physiological functions of each major cluster were based on a normal baseline (as opposed to mixing based on 8 samples as previously), and then the comparison between normal male and non-DM/DMED patients in each subcluster were made. (2) Pericyte, which was named as "SMC3" in the last version, was identified according to the suggestion of

reviewer #1. They were found around small blood vessels in the CC tissue. (3) The nomenclature of MRF was abandoned, and this cluster of cells was analyzed after integration with fibroblasts from other multiple tissues, and a more standardized nomenclature (“MRF” was replaced by “Fibroblast, FB”). Under this classification criteria, fibroblasts are divided into 3 major clusters (PI16+, APOC1+, and COMP+) (Buechler et al. 2021), and further into 6 subcluster based on some CCFB specific markers including APOC1+/PTCHD1+, APOC1+/PPP1R14A+, PI16+/FMO2+, PI16+/BMP7+, COMP+/KERA+ and COMP+/MFAP5+ FB. (4) Although there is some evidence from cell experiments *in vitro* and *ex vivo*, we were more cautious with the results of “fibroblast differentiation into myofibroblasts” and “TGFb induced EndoMT”, and describe only the experimental phenomena. (5) IHC staining of YAP/p-YAP with SMA protein to observe the change of YAP signaling in CC SMC (corpus cavernosal trabecular smooth muscle cells) were added. (6) We validated CellChat's predictions from expression and function two aspects *in vitro*.

Q1

Diabetic erectile dysfunction is a combination of microvascular, macrovascular, endocrine, and neurological disorders, and the authors seem to have described most of the etiologies. However, whether the angiogenic factors, neuroregenerative factors, and endocrine-related signaling pathways were also altered? Especially in normal and DMED patient.

Response: Thank you for your suggestion, we have deleted the cell-cell communication base on the mix of three types of samples, and added a comparison between normal and DMED to analyze what signaling may cause the pathological injury in DMED CC (Figure R3-1, it is also added as the new Supplementary Figure 2 in the revised manuscript). The results showed that most signaling in DMED CC were increased, it may be related to the regulation of feedback under pathological injury stimulus. However, there are two interesting observations that are worth noting. First, the overall IGF1 signaling was generally decreased in patients, but it was specially increased in intensity between FB and EC cluster, and between FB and SMC cluster, and this was consistent with the results observed by IHC in our last version manuscript. In addition, KIT signals were significantly decreased or even disappeared in DMED patients. In recently functional experiments, we found SCF (the ligand of KIT) treatment facilitates endothelial cell migration, tube formation, vascular bud length, and NO release, and inhibits ET release, all of these effects were beneficial for damage repair of CC tissue and relaxation of smooth muscle (Figure R3-2).

Figure R3-1 (Supplementary Figure 2 in the revised manuscript). Cell-cell communication network within the corpora cavernosa microenvironment in normal and pathological state. (a) The barplot showed the total interactions count and strength among normal, non-DM and DMED CC. (b) The networks showed the total interactions count and strength with 7 major clusters among normal, non-DM and DMED CC. (c) The heatmap showed the outcoming signaling patterns between normal and DMED CC, the barplot on the right of heatmap indicated the outcoming strength. (d) Circle plots (by CellChat analysis) depict the differential strength (right) of specific signaling in the cell-cell communication network between normal and DMED CC. Red or blue edges represent increased or decreased signaling in normal state.

Figure R3-2 (parts of Supplementary Figure 4 and 5 in revised manuscript). CCEC tube formation (a), vascular bud length (b), migration (c), endothelial NO release (d) and endothelin release (e) change after single factor stimulation. The statistical analysis was made by ANOVA with Tukey's multiple comparisons test; the confidence interval is 95%.

Q2

Diabetic ED is known to be associated with severe structural damage, if possible, please add the cell type (cluster) ratio for the eight major cavernosal cell types from normal and ED patients.

Response: Thank you for your advice. We checked the ratio of the 7 major cavernosal cell types (ENF and MFB were classified into EC and SMC cluster respectively) and 11 major cluster (a more detail classification approach where CC and nearby vessels were divided) from normal and ED patients. However, according to scRNA-seq results, more SMC cell were captured in DMED samples (Figure R3-3a and R3-3b). It was inconsistent with what we observed in HE, Masson and IHC staining (DMED CC often contain less EC and SMC, but more FB). We hypothesize that this difference may be caused by three factors. First, the normal CC tissue were the whole circumferential cross section which obtained from penile carcinoma resection, while the DMED CC tissue were obtained from artificial cavernous body implantation, these samples were closer to albuginea and less abundant in the vascular-rich part (Figure R3-3c and R3-3d). Second, the CC smooth muscle bundles in DMED patients often atrophy, it is more difficult to dissociate the thicker smooth muscle bundles of normal CC into single cells and captured. The third, the ED samples showed more severe fibrosis and fewer fibroblasts within the extracellular matrix proteins were released under the same dissociation conditions. In order not to mislead the readers with this contradictory result, we suggest not to show this part of the results, but we would add a relative discussion in the revised manuscript, and we suggest that the damage of CC tissue structure should be evaluated based on morphological staining.

Figure R3-3. (a-b) the ratio of the 7 major cavernosal cell types and 11 major cluster from normal and ED patients. (c-d) H&E staining and gross specimen of CC cross section, the region inside the rectangular contains more vessels.

Q3

Please provide more detailed information for subjects included in the study.

- IIEF EF or IIEF-5 scores for all subjects
- Please provide previous ED treatment modality, such as oral PDE5I or ICI for 5 patients with ED
- Two patients with diabetic ED: please provide data for HbA1c, anti-glycemic medication, etc
- Three patients with non-diabetic organic ED; what is exact cause for ED? More detailed information, such as vascular risk factors or neurological causes, are needed.

Response: Thank you for your suggestion. In the last version, the description of clinical information was too simple, which may have caused some confusion. In fact, the 8 patients enrolled in this study were screened carefully, and we tried to exclude those who were older than 55 years old. Although most penile cancer patients are older than this age, increasing age also leads to more chronic disease and other confounding factors. So, we chose 55 years old as a balance that allowed us to get enough samples, but also control for age-related interference. Other patients who had other chronic diseases or bad lifestyle habits that seriously affected their health were also excluded. For the 2 DMED patients (Patients with DMED often cannot choose surgical treatment because they are prone to post-operative infections and poor wound healing, resulting in difficulty in obtaining samples), both of them had been diagnosed with type 1 diabetes for more than 10 years. For non-DM patients, we also tried to select patients with relatively consistent symptoms according to ICI tests, they all had

both inadequate blood supply and increased venous return (Figure R3-4a). **A more detailed clinical information of the 8 patients was added as Supplement Table 1 in the revised manuscript.**

a

Sample ID	ED duration (Year)	IIEF-5 score	NPTR Effective erection event/Erectile event	Tip/Base maximum average hardness	ICI PSV (left/right) cm/s	ICI EDV (left/right) cm/s	ICI RI (left/right)	ICI erection hardness
non-DM_1	5	7	0/2	Tip/Base maximum average hardness 47% and 64%, swelling 19% and 22%, lasting 2 min.	30/28	10/8	0.66/0.71	Grade 2
non-DM_2	4	9	0/8	Tip/Base maximum average hardness 59% and 41%, swelling 31% and 31%, lasting 13 min.	24/25	7/6	0.70/0.76	Grade 3
non-DM_3	3	8	0/10	Tip/Base maximum average hardness 62% and 60%, swelling 29% and 30%, lasting 7 min.	18/27	4/0	0.77/1	Grade 3
Diabetes_1	3	13	0/1	Tip/Base maximum average hardness 45% and 40%, swelling 19% and 22%, lasting 4 min.	29/47	9/15	0.68/0.68	Grade 2-3
Diabetes_2	5	11	0/1	Tip/Base maximum average hardness 52% and 34%, swelling 37% and 21%, lasting 5 min.	27/28	5/6	0.81/0.80	Grade 2

b

Figure R3-4. (a) The clinical information of the 8 patients in this study. (b) The NPTR results of the 8 patients. Normal reference value: penis dilation hardness $\geq 60\%$ and duration $\geq 10\text{min}$. Free testosterone $\geq 0.0865\text{ng/mL}$. PSV $\geq 30\text{cm/s}$, EDV $\leq 3\text{cm/s}$, RI ≥ 0.8 .

Q4

The characteristics of organic ED are somewhat not homogenous, i.e., 2 patients with diabetic ED and 3 patients with non-diabetic organic ED. I think this is a major limitation of single cell transcriptome analysis used in this study.

Response: we agree that the characteristics of organic ED are somewhat not homogenous. In fact, in most parts of the paper, DMED and non-DM were analyzed as two independent groups and compared with normal people respectively. We never mix all ED patients together. In addition, the

heterogeneity of DMED and non-DM patients was just part of our analysis results, i.e., the different changes of EC and its mitochondria, which contributed to the in-depth understanding of ED caused by various etiologies.

- Armulik A, Genové G, Betsholtz C. 2011. Pericytes: developmental, physiological, and pathological perspectives, problems, and promises. *Dev Cell* **21**: 193–215.
- Buechler MB, Pradhan RN, Krishnamurthy AT, Cox C, Calviello AK, Wang AW, Yang YA, Tam L, Caothien R, Roose-Girma M et al. 2021. Cross-tissue organization of the fibroblast lineage. *Nature* **593**: 575–579.
- De Micheli AJ, Spector JA, Elemento O, Cosgrove BD. 2020. A reference single-cell transcriptomic atlas of human skeletal muscle tissue reveals bifurcated muscle stem cell populations. *Skeletal muscle* **10**: 19.
- Deng CC, Hu YF, Zhu DH, Cheng Q, Gu JJ, Feng QL, Zhang LX, Xu YP, Wang D, Rong Z et al. 2021. Single-cell RNA-seq reveals fibroblast heterogeneity and increased mesenchymal fibroblasts in human fibrotic skin diseases. *Nature communications* **12**: 3709.
- Habermann AC, Gutierrez AJ, Bui LT, Yahn SL, Winters NI, Calvi CL, Peter L, Chung MI, Taylor CJ, Jetter C et al. 2020. Single-cell RNA sequencing reveals profibrotic roles of distinct epithelial and mesenchymal lineages in pulmonary fibrosis. *Science advances* **6**: eaba1972.
- Kalucka J, de Rooij L, Goveia J, Rohlenova K, Dumas SJ, Meta E, Conchinha NV, Taverna F, Teuwen LA, Veys K et al. 2020. Single-Cell Transcriptome Atlas of Murine Endothelial Cells. *Cell* **180**: 764–779 e720.
- Kinchen J, Chen HH, Parikh K, Antanaviciute A, Jagielowicz M, Fawcner-Corbett D, Ashley N, Cubitt L, Mellado-Gomez E, Attar M et al. 2018. Structural Remodeling of the Human Colonic Mesenchyme in Inflammatory Bowel Disease. *Cell* **175**: 372–386 e317.
- Kuppe C, Ibrahim MM, Kranz J, Zhang X, Ziegler S, Perales-Patón J, Jansen J, Reimer KC, Smith JR, Dobie R et al. 2021. Decoding myofibroblast origins in human kidney fibrosis. *Nature* **589**: 281–286.
- Li J, Yao M, Zhu X, Li Q, He J, Chen L, Wang W, Zhu C, Shen T, Cao R et al. 2019. YAP-Induced Endothelial-Mesenchymal Transition in Oral Submucous Fibrosis. *Journal of dental research* **98**: 920–929.
- Mäe MA, He L, Nordling S, Vazquez-Liebanas E, Nahar K, Jung B, Li X, Tan BC, Chin Foo J, Cazenave-Gassiot A et al. 2021. Single-Cell Analysis of Blood-Brain Barrier Response to Pericyte Loss. *Circulation research* **128**: e46–e62.
- Ramachandran P, Dobie R, Wilson-Kanamori JR, Dora EF, Henderson BEP, Luu NT, Portman JR, Matchett KP, Brice M, Marwick JA et al. 2019. Resolving the fibrotic niche of human liver cirrhosis at single-cell level. *Nature* **575**: 512–518.
- Rao M, Wang X, Guo G, Wang L, Chen S, Yin P, Chen K, Chen L, Zhang Z, Chen X et al. 2021. Resolving the intertwining of inflammation and fibrosis in human heart failure at single-cell level. *Basic Res Cardiol* **116**: 55.
- Reyfman PA, Walter JM, Joshi N, Anekalla KR, McQuattie-Pimentel AC, Chiu S, Fernandez R, Akbarpour M, Chen CI, Ren Z et al. 2019. Single-Cell Transcriptomic

- Analysis of Human Lung Provides Insights into the Pathobiology of Pulmonary Fibrosis. *American journal of respiratory and critical care medicine* **199**: 1517–1536.
- Savorani C, Malinverno M, Seccia R, Maderna C, Giannotta M, Terreran L, Mastrapasqua E, Campaner S, Dejana E, Giampietro C. 2021. A dual role of YAP in driving TGFbeta-mediated endothelial-to-mesenchymal transition. *J Cell Sci* **134**.
- Terentes-Printzios D, Ioakeimidis N, Rokkas K, Vlachopoulos C. 2021. Interactions between erectile dysfunction, cardiovascular disease and cardiovascular drugs. *Nat Rev Cardiol* doi:10.1038/s41569-021-00593-6.
- Theocharidis G, Thomas BE, Sarkar D, Mumme HL, Pilcher WJR, Dwivedi B, Sandoval-Schaefer T, Sîrbulescu RF, Kafanas A, Mezghani I et al. 2022. Single cell transcriptomic landscape of diabetic foot ulcers. *Nature communications* **13**: 181.
- Vanlandewijck M, He L, Mäe MA, Andrae J, Ando K, Del Gaudio F, Nahar K, Lebouvier T, Laviña B, Gouveia L et al. 2018. A molecular atlas of cell types and zonation in the brain vasculature. *Nature* **554**: 475–480.
- Zhang X, Lan Y, Xu J, Quan F, Zhao E, Deng C, Luo T, Xu L, Liao G, Yan M et al. 2019. CellMarker: a manually curated resource of cell markers in human and mouse. *Nucleic acids research* **47**: D721–d728.

REVIEWER COMMENTS

Reviewer #1 (Remarks to the Author):

The revised manuscript is substantially improved and the authors have adequately addressed many of my earlier concerns, especially with respect to the MRF and pericytic populations. However, I still have a couple of more minor issues which require attention.

In the first rebuttal section, not sure what is meant by "differentiation efficiency was low and about only 5~10% CC fibroblast cells were induced differentiation successfully in vitro". Was this carried out in clonal assays, so only about 5-10% of the clones exhibited tri-lineage differentiation capability. I noticed this data is not included in the revised manuscript, so this is a minor concern.

Major comments 4 and 5 – the generic myocyte and muscle terms as used are confusing. Please be more specific, are they transcriptional signatures reflective of striated muscle or smooth muscle. Supplementary Figure 11e contains a number of muscle and non-muscle markers, if you are studying bona fide striated muscle, then myogenic factors such as MYOD, MYF5, etc. should be detectable and not transcripts reflective of smooth muscle, i.e., MYOCD. Furthermore, the findings that cultured "CC fibroblasts" differentiate into myocytes needs further experimental validation. More recent lineage tracing studies in murine models have shown that fibroblasts do not contribute to the myogenic lineage. This line of investigation doesn't add much to the overall study, so it could be removed.

Minor comment 8 – To make a convincing case for the endothelial fibroblast, this putative cell type would need to be quantified within your samples. The weak potential overlapping stain of one example shown in 14a does not provide solid support. This is an issue we have grappled with for years in analyzing these cell types, and their close proximity makes their resolution challenging, unless they appear at high frequency. Without compelling experimental support, I would remove this component.

Reviewer #2 (Remarks to the Author):

Excellent revision.

The authors have provided functional (e.g. nuclear Yap1 localization) that independently validates the results from their detailed informatics, and have clarified the numbers of human subjects that contributed to the foundational atlas they have developed.

My only remaining minor comment is that, for the benefit of the Journal's broad readership, that they comment on the following: while extremely powerful, the groupings used to define the specific cell types and subclusters identified in sc-RNA-seq data is somewhat arbitrary - albeit supported by their IF data. In their re-analysis, the regrouping from 8 to 7 major cavernosal cell types reflects this fact. Moreover, as another example, on line 266 while the authors point to the tendon-like cell population (FB4) based upon SCX / Scleraxis expression, Scx+ cells have been identified as important in progenitors for cardiac valve morphogenesis. This does not reduce the significance of the authors' work at all, but highlights the current challenges of defining mesenchymal cell subsets by transcriptomic technologies (vs. cell surface CD markers or chromatin epigenetic / architecture as in immunocytes). As data accrues concerning the mesenchymal cell lineage, phenotypic modulation, with consensus definitions by sc-RNA-Seq and spatial transcriptomics that encompass SMC heterogeneity, this will improve.

Agata et al, Scleraxis is required for cell lineage differentiation and extracellular matrix remodeling during murine heart valve formation in vivo. Circ Res. 2008 Oct 24;103(9):948-56.

Reviewer #3 (Remarks to the Author):

The authors have revised the manuscript based on my comments and I have no additional comments.

Reviewer #1 (Remarks to the Author):

The revised manuscript is substantially improved and the authors have adequately addressed many of my earlier concerns, especially with respect to the MRF and pericytic populations. However, I still have a couple of more minor issues which require attention.

In the first rebuttal section, not sure what is meant by "differentiation efficiency was low and about only 5~10% CC fibroblast cells were induced differentiation successfully in vitro";. Was this carried out in clonal assays, so only about 5-10% of the clones exhibited tri-lineage differentiation capability. I noticed this data is not included in the revised manuscript, so this is a minor concern.

Response: Thank you for your concern. We showed this data in the first rebuttal section because we want to explain why we named the FB cluster as Mesenchymal regulated fibroblast (MRF).

According to Mesenchymal and Tissue Stem Cell Committee of the International Society for Cellular Therapy proposes minimal criteria to define human MSC. First, MSC must be plastic-adherent when maintained in standard culture conditions. Second, MSC must express CD105, CD73 and CD90, and lack expression of CD45, CD34, CD14 or CD11b, CD79alpha or CD19 and HLA-DR surface molecules. Third, MSC must differentiate to osteoblasts, adipocytes and chondroblasts in vitro.

In the last rebuttal version, the result of ossification and adipogenesis showed a positive region where CC fibroblast was successfully induced differentiation. However, most CC fibroblast cells remain unchanged, and the result "only about 5-10% of the clones exhibited tri-lineage differentiation capability" was based on the ratio between positive and negative region area (Alizarin red staining and oil red staining).

Because we have removed all contents and concepts associated with mesenchymal cells in the revised manuscript (the cluster name MRF was replaced by FB), so its related data were also deleted.

Figure1: Alizarin red staining of CC FB under differentiate to osteoblasts inducing *in vitro*. Only small part of CC FB were induced successfully.

Major comments 4 and 5 – the generic myocyte and muscle terms as used are confusing. Please be more specific, are they transcriptional signatures reflective of striated muscle or smooth muscle. Supplementary Figure 11e contains a number of muscle and non-muscle markers, if you are studying bona fide striated muscle, then myogenic factors such as MYOD, MYF5, etc. should be detectable and not transcripts reflective of smooth muscle, i.e., MYOCD. Furthermore, the findings that cultured “CC fibroblasts” differentiate into myocytes needs further experimental validation. More recent lineage tracing studies in murine models have shown that fibroblasts do not contribute to the myogenic lineage. This line of investigation doesn’ t add much to the overall study, so it could be removed.

Response: Thank you for your advice, we have further removed any conclusion about FB differentiate into myocytes, but the description of the result that FB treated with ICG-001 or SKL was remained. Because the activation of WNT was a significant phenotype in DMED FB, and we wanted to exhibit how the transcription of CCFB changes after WNT activation *in vitro*, it could contain the upregulation of some muscle related genes, but we never emphasized that this result represented FB was induced into the myogenic lineage or myofibroblasts in the revised manuscript. This result only represents a phenotypic transformation in disease state rather than lineage differentiation. So, we did not distinguish this up-regulated muscle related genes belong to striated muscle or smooth muscle lineage in Supplementary Figure 11e. Another reason is the myogenic factors you advised such as MYOD, MYF5 and MYOCD were not

found in the DEGs after SKL treatment (Supplementary table 10).

We agree with your concern that cultured CCFB differentiate into myocytes needs further experimental validation, so we have removed all conclusion about this in the last revised manuscript. We found that the GO pathway "muscle structure development" shown in the volcano plot probably caused this confusion (because the word "development" implies differentiation), so we replace this term with "muscle contractile" in the revised manuscript (it could be found in Figure 3g and Supplementary Figure 11d) and also found a similar result (the up-regulated genes was much more than down-regulated genes in SKL treatment group), it is worthy note that the up-regulated genes in this pathway not only contained the executor of muscle contraction such as ACTA2 and CACNA1H, but also contained some regulatory signal molecules such as EDN1. Neither the constriction of the supplying artery muscle nor of the CC sinus is conducive to penile erection. So, we thought this result represented a pathological state of CCFB (although themselves did not differentiate into myocytes) in ED patients.

Another your concern "More recent lineage tracing studies in murine models have shown that fibroblasts do not contribute to the myogenic lineage". We found that the current research may be controversial. In 2021, Joana Esteves et al. reported the unexpected contribution of fibroblasts to muscle lineage, and they also show that BMP signalling regulates a fibroblast-to-myoblast conversion both in vivo and in vitro (Esteves de Lima J, et al. Nat Commun. 2021 Jun 22;12(1):3851). However, in the new revised manuscript, we have deleted all the contents related to CCFB into myogenic lineage differentiation, so these uncertainties will not affect this study. Thank you again!

Minor comment 8 – To make a convincing case for the endothelial fibroblast, this putative cell type would need to be quantified within your samples. The weak potential overlapping stain of one example shown in 14a does not provide solid support. This is an issue we have grappled with for years in analyzing these cell types, and their close proximity makes their resolution challenging, unless they appear at high frequency. Without compelling experimental support, I would remove this component.

Response: Thank you for your comment. We have benefited a lot from your advice, because we did not carefully consider the situation of double-cell cognition before. If there is an opportunity in future, I will be very honored to have in-depth discussion with you and try to solve this problem. It maybe depends on analysis of FB markers of ICC staining after isolating CD31 positive cells from ED CC tissue with FACS.

Here, we try to provide other related data. The first is we found the frequency of endothelial fibroblast was not very low (in order to observe the localization of membrane proteins and cytoplasmic proteins clearly, we adopted the picture of magnified field in the Figure R1-14a of last rebuttal), we now show two other

regions of DMED CC tissue in a 20x objective, and more CD31/LUM double positive cells could be found. The second is we noticed that EC and FB may not show close proximity in most part of CC tissue, for example that in the Figure R1-14a of last rebuttal, it is noted that no other cells are present adjacent to the continuous EC. In addition, in the SEM image, extracellular matrix fibers, but not fibroblasts, were exposed at the endothelial disappeared regions (fibroblasts are usually in a more inner layer), indicating these two cell types were not always combined with each other tightly.

Figure2

In summary, thank you very much for taking the time to review our study and putting forward many constructive suggestions, which we believe will ultimately benefit clinical patients.

Reviewer #2 (Remarks to the Author):

Excellent revision.

The authors have provided functional (e.g. nuclear Yap1 localization) that independently validates the results from their detailed informatics, and have clarified the numbers of human subjects that contributed to the foundational atlas they have developed.

My only remaining minor comment is that, for the benefit of the Journal's broad readership, that they comment on the following: while extremely powerful, the groupings used to define the specific cell types and subclusters identified in sc-RNA-seq data is somewhat arbitrary - albeit supported by their IF data. In their re-analysis, the regrouping from 8 to 7 major cavernosal cell types reflects this fact. Moreover, as another example, on line 266 while the authors point to the tendon-like cell population (FB4) based upon SCX / Scleraxis expression, Scx+ cells have been identified as important in progenitors for cardiac valve morphogenesis. This does not reduce the significance of the authors' work at all, but highlights the current challenges of defining mesenchymal cell subsets by transcriptomic technologies (vs. cell surface CD markers or chromatin epigenetic / architecture as in immunocytes). As data accrues concerning the mesenchymal cell lineage, phenotypic modulation, with consensus definitions by sc-RNA-Seq and spatial transcriptomics that encompass SMC heterogeneity, this will improve.

Agata et al, Scleraxis is required for cell lineage differentiation and extracellular matrix remodeling during murine heart valve formation in vivo. *Circ Res.* 2008 Oct 24;103(9):948-56.

Response: Thank you for your review and suggestions which improved our research a lot. We fully understand your concerns, in fact, we have also been exploring appropriate classification criteria of the CC cells, so in the revised manuscript, we have adjusted the major clusters, such as endothelial fibroblasts are classified into endothelial cell cluster. Because in the lineage differentiation, almost all of the study showed that this group come from endothelial cell differentiation, rather than fibroblast lineage or other separate developmental pathways. So, we think the new classification standard is more scientific.

In addition, your suggestion is constructive, and the sequencing analysis of spatial transcriptome combined with multiomics analysis is the focus of our further work, which can solve many limitations left in this study, and we emphasized this point in the new revised manuscript. Furthermore, we did find a lot of similarities between the CC and the heart such as you noticed Scx+ cells, and that may partly explain why ED are often followed by cardiovascular events. However, we think this is not necessarily a bad thing. Because in the basic research of andrology, especially in ED, the resources and history are far less than cardiovascular systems. These results also suggest that future studies on the pathogenesis and treatment of ED should learn more from existing cases

in the cardiovascular field, which may facilitate the rapid development of andrology.

Finally, thank you for your recognition of our revision work, and sincerely wish you a smooth work and a happy life!

Reviewer #3 (Remarks to the Author):

The authors have revised the manuscript based on my comments and I have no additional comments.

REVIEWERS' COMMENTS

Reviewer #1 (Remarks to the Author):

The authors have adequately addressed my concerns.